# The Association between the Atherogenic Index of Plasma and Cardiometabolic Risk Factors: A Review

**DOI:** 10.3390/healthcare11070966

**Published:** 2023-03-28

**Authors:** Beatrice Lioy, Richard James Webb, Farzad Amirabdollahian

**Affiliations:** 1Faculty of Science and Engineering, Manchester Metropolitan University, Manchester M15 6BH, UK; 2School of Health and Sport Sciences, Liverpool Hope University, Hope Park Campus, Liverpool L16 9JD, UK; 3Faculty of Education, Health and Wellbeing, The University of Wolverhampton, Millennium City Building, Wulfruna St, Wolverhampton WV1 1LY, UK

**Keywords:** atherogenic index of plasma, cardiometabolic risk, metabolic syndrome, high-density lipoprotein cholesterol, triglycerides, waist circumference, low-density lipoprotein cholesterol, hypertension, body mass index, insulin resistance

## Abstract

Background: Metabolic syndrome (MetS) is a condition caused by a combination of cardiometabolic risk factors (CMR). MetS leads to type 2 diabetes mellitus (T2DM) and cardiovascular disease (CVD), both of which place a burden on not only the patients but also the healthcare system. Diagnostic criteria for MetS vary, and there is no universal tool to detect it. Recently, many studies have found positive associations between the atherogenic index of plasma (AIP) and some CMR factors. Therefore, a comprehensive review was needed to recapitulate these studies and qualitatively estimate the likelihood of AIP being associated with CMR. We aimed to review and summarise observational data on AIP and CMR factors and verify their association. Materials and Methods: A review of observational studies was conducted by searching “atherogenic index of plasma” in PubMed, One Search, and the Cochrane library. A total of 2068 articles were screened, and 32 were included after excluding paediatric, non-human and interventional studies, and those carried out on cohorts with conditions unrelated to MetS or on lipid-lowering medication. The Newcastle-Ottawa scale was used to assess their quality. Results: Most studies that reported high waist circumference (WC), triglycerides (TG), insulin resistance (IR) and low high-density lipoprotein cholesterol (HDL-C) concentration, also reported high AIP. Few studies investigated blood pressure (BP) and some discrepancies existed between their results. Conclusion: AIP may be associated with WC, TG, IR, and HDL-C. It is unclear if AIP is associated with BP. The current study’s results should be used to inform futureward a meta-analysis to be seen quantitatively. It is also recommended that more cohort studies stratified by gender and ethnicity be performed to ascertain if AIP can predict MetS before it manifests.

## 1. Introduction

“Cardiometabolic risk” (CMR) is an umbrella term that covers a variety of risk factors such as dyslipidaemia, central obesity, hypertension, and impaired glucose metabolism [1]. A combination of CMR factors leads to metabolic syndrome (MetS), which leads to type 2 diabetes mellitus (T2DM) and cardiovascular disease (CVD) [2]. Although differences in MetS definitions exist, as shown in Table 1, this paper adopted the definition by the International Diabetes Federation (IDF) due to its stricter criteria, as this might be more useful for prevention. CMR typically arises because of prolonged sedentary behaviour [3] and poor diet, characterised by ultra-processed, high-fat, high-sodium foods [4]. Excessive energy and fat intake in the absence of physical activity naturally contribute to the aetiology of obesity, and similarly, the imbalance between high-density cholesterol (HDL-C) and low-density cholesterol (LDL-C) contributes to the aetiology of dyslipidaemia. The IDF considers waist circumference (WC) to be an essential criterion for MetS diagnosis. Excessive triglycerides interfere with the normal functioning of the reverse cholesterol pathway, which in turn contributes towards atherosclerosis, insulin resistance (IR), and inflammation. T2DM is an independent risk factor for CVD, a clinical condition affecting around 32.2% of T2DM patients [5]. This is particularly concerning since CVD is the leading cause of mortality globally, accounting for around 32% of all death [6]. Furthermore, it has been estimated that the prevalence of chronic complications of diabetes ranged from 8.1% to 41.5% for retinopathy, 21% to 22% for albuminuria, 6.7% to 46.3% for nephropathy, and 21.9% to 60% for neuropathy [7]. Moreover, it is well established that both CVD and diabetes are associated with a worsened quality of life and depression [8,9,10,11].

Therefore, due to the burden of these conditions on both the patients and healthcare systems worldwide, a cheap, reliable, and an easily available tool is needed to routinely identify high-risk cases. Several studies have investigated the predictive ability of the atherogenic index of plasma (AIP), calculated as log^10^ (TG/HDL-C), which was introduced by Dobiásová et al. in 2000 [12]. However, to date, there is no comprehensive review on this topic. To the best of the authors’ knowledge, only one review of six studies exists; it is over 15 years old and does not represent the current literature [13]. This is important because over recent years, many studies have investigated AIP and numerous have revealed a positive association between AIP and cardiometabolic risk factors [14,15,16,17,18]. Moreover, they found that an AIP value of under 0.11 is associated with a low risk of CVD, and values between 0.11 to 0.21 and upper than 0.21 are associated with intermediate and increased risks, respectively [19,20]. These findings, together with the need to identify people at risk of CVD and T2DM, warrant more research. In light of this, we aimed to review the body of literature to summarise observational data interlinking AIP with CMR factors and verify their association.

**Table 1 healthcare-11-00966-t001:** Diagnostic biochemical and anthropometric criteria defining MetS according to major organisations, including NCEP [21], WHO [22], IDF [23], EGSIR [24], AACE [25].

CMR Factors	NCEP (National Cholesterol Education Programme) ATP3	WHO (World Health Organisation)	IDF (International Diabetes Federation)	EGSIR (European Group for the Study of Insulin Resistance Criteria)	(AACE) American Association of Clinical Endocrinology
Based on	(Expert panel on detection, evaluation, and treatment of high blood cholesterol in adults, 2001)	(Alberti and Zimmet, 1998)	(Alberti et al., 2005)	(Balkau and Charles, 1999)	(Einhorn et al., 2003)
IR	Blood glucose > 5.6 mmol/L (100 mg/dL) or drug treatment for elevated blood glucose	Blood glucose > 6.1 mmol/L (110 mg/dL), 2 h glucose > 7.8 mmol (140 mg/dL)	Blood glucose > 5.6 mmol/L (100 mg/dL) or diagnosed diabetes	Insulin levels > 75th percentile of non-diabetic patients; Blood glucose 110 mg/dL or greater	
HDL-C	<1.0 mmol/L (40 mg/dL) in men, <1.3 mmol/L (50 mg/dL) in women or drug treatment for low HDL-C	<0.9 mmol/L (35 mg/dL) in men, <1.0 mmol/L (40 mg/dL) in women	<1.0 mmol/L (40 mg/dL) in men, <1.3 mmol/L (50 mg/dL) in women or drug treatment for low HDL-C	<39 mg/dL in men or women	<40 mg/dL in men and <50 mg/dL in women
TG	>1.7 mmol/L (150 mg/dL) or drug treatment for elevated triglycerides	>1.7 mmol/L (150 mg/dL)	>1.7 mmol/L (150 mg/dL) or drug treatment for elevated triglycerides	150 mg/dL or greater	150 mg/dL or greater
WC	>102 cm (men) or >88 cm (women)		Europeans: >94 cm (men) or >80 cm (women) South Asians and Chinese: >90 cm (men) >80 cm (women) Japanese: >85 cm (men) >90 cm (women)	94 cm or greater in men, 80 cm or greater in women	
WHpR		>0.9 (men) or >0.85 (women)			
BMI		>30 kg/m^2^			25 kg/m^2^ or greater
HPT	>130/85 mmHg or drug treatment for hypertension	>140/90 mmHg	>130/85 mmHg or drug treatment for hypertension	140/90 mmHg or greater or taking antihypertensive drugs	130/85 mmHg or greater
MetS diagnostic criteria	3 or more factors	IR + 2 or more other factors	WC + 2 or more other factors	IR + 2 or more other factors	IGT + 2 or more factors

CMR = cardiometabolic risk; IR = insulin resistance; HDL-C = high density lipoprotein cholesterol; TG = triglycerides; WC = waist circumference; WHpR = waist-to-hip ratio; BMI = body mass index; HPT = hypertension; MetS = metabolic syndrome; IGT = impaired glucose test.

## 2. Materials and Methods

### 2.1. Study Design and Rationale

Although our overall study design followed a systematic approach, we acknowledge that it did not follow all the rigorous principles recommended by the Cochrane review guidelines. Therefore, given these shortcomings, we decided to present our findings within the framework of a systematically conducted narrative review.

### 2.2. Search Strategy

A systematic search of PubMed, Cochrane library, and the Liverpool Hope University online library named “One Search” was carried out according to the Preferred Reporting Items for Systematic Review and Meta-Analysis protocols. The terms “atherogenic index of plasma” were searched in each database. No publication date limits were imposed.

### 2.3. Study Selection and Inclusion/Exclusion Criteria

All studies found were systematically screened three times by one reviewer (BL). The initial screening involved titles and abstracts only, while the second and third required scrutiny of the studies’ methodology, baseline data, and inclusion/exclusion criteria. A PRISMA flowchart was made to illustrate this process (Figure 1). A second reviewer (FA) was involved when there was doubt regarding a study’s inclusion. 

Inclusion criteria included: (1) observational studies (including cohort studies, case–control studies, retrospective/prospective studies); (2) adult participants (>18 years); (3) general population with or without MetS components (including apparently healthy, with CVD/T2DM/hypertension/dyslipidaemia); (4) studies that calculated AIP and measured at least one blood biomarker related to MetS (including HDL-C, LDL-C, TC, TG, SBP/DBP, glucose, insulin) as baseline data. 

Exclusion criteria included: (1) non-observational studies (including any study that had any intervention, such as randomised control trials involving bariatric surgery, food or medicine consumption, Ramadhan, ketogenic diet, or physical exercise); (2) non-human participants (performed on mouse, rabbit, chicken, or any other non-human animal); (3) paediatric (<18 years of age); (4) performed on cohorts exclusively selected for having a condition unrelated or not directly related to metabolic syndrome (including renal disease, type 1 diabetes, human-immunodeficiency-virus, polycystic ovary syndrome, malignancy, pregnancy, genetic polymorphisms, sleep and mood disorders, autoimmune conditions, inflammatory and infectious conditions, familial hypercholesterolemia); (5) if the study reported lipid, blood pressure, or glucose-lowering medications use among participants (including statins, metformin, pioglitazone, insulin), or if the study did not confirm the absence of such medications; (6) if AIP was not expressed as a mean (excluded if expressed as a range, for example); (7) if AIP and at least one CMR-related biomarker were not calculated/measured and shown as baseline data (for example, if AIP was only used to calculate a correlation coefficient); and (8) if search results consisted within newsletter articles or correspondence that did not involve a new study with new usable data.

Research manuscripts reporting large datasets that are deposited in a publicly available database were considered; however, they needed to specify where the data have been deposited and provide the relevant accession numbers. Intervention studies involving humans, and other studies that require ethical approval, must have listed the authority that provided approval and the corresponding ethical approval code.

### 2.4. Quality Assessment of Included Studies

The quality and scientific rigour of all studies included in the current review were assessed using the Newcastle-Ottawa quality assessment scales for cohort, case–control, and cross-sectional (adapted scale) studies. Such a tool is recommended by the Cochrane collaboration as its use may reduce assessment bias and the chances of conclusions being made based on low-quality studies that fail to show transparency regarding methodology and data handling [26]. Our systematic approach in review aimed to synthesise high-quality studies to provide stronger evidence, hence, this step was considered of the essence. Table 2 shows the scores obtained by each study. 

The scales used required that the study was to be scored based on three main criteria: selection, comparability, and exposure/outcome. In the case of cross-sectional studies, the selection focused on sampling strategies, comments and/or characteristics of non-responders, and ascertainment of exposure. Comparability focused on whether the study controlled for main and additional factors. Outcome focused on how outcomes were assessed and the adequacy of statistical tests. For case–control studies, the scale used was very similar in terms of comparability and exposure (exposure as the equivalent of the outcome but for case–control studies). Key differences were in the selection section, which focused on the adequacy of “case” and “control” definitions, where the control cases came from, and how well and objectively the cases were being represented. For both scales, a maximum number of stars could be given, where more stars indicated a higher score. After careful research, it was determined that there is no universally accepted threshold of stars to determine if a study is of high or low quality. In the absence of official guidelines, the authors made the decision to use the following criteria: “5/7” and “7/9” being “high quality” studies”, and anything below those scores being of “low quality” or, as some label, at “high risk of bias”.

## 3. Results

The results of our current review are presented in Table 3, Table 4 and Table 5 showing the association between the AIP and cardiometabolic risk: Table 3 shows the association between anthropometric measurements and AIP; Table 4 shows blood lipid measurements and AIP; Table 5 shows insulin resistance measurements, T2DM prevalence among participants, and AIP; and Table 6 shows BP measurements, HPT prevalence among participants, and AIP. To facilitate further reading, every instance in which the measured data met the thresholds for MetS diagnosis was shown, as highlighted in Table 1. Moreover, the word “high” was placed next to values exceeding an upper threshold, and the word “low” was placed next to HDL-C values that did not meet the minimum threshold. AIP values were also coded as “high” (>0.21), “medium” (0.11–0.21), and “low” (<0.11) to indicate the severity of risk as highlighted in previous studies. Despite anthropometric measurements not being part of the chosen IDF criteria, these were reported in the tables to form part of the discussion. BMI was deemed “high” when >30 kg/m^2^, and “medium” when 25–30 kg/m^2^. When gender was not specified in a study, codes were followed by a question mark as it was not possible to make a firm judgement based on the reported index. 

No study explicitly stated the research question as their primary outcome due to the nature of observational studies. Standard deviation was shown next to measured outcomes where available, which was not always the case. The same applied to *p*-values, which were only noted if given and if significant. Most studies stratified their cohorts based on criteria such as gender and presence or absence of disease, and this stratification was maintained in the current study’s results tables for discussion purposes. Among all thirty-one studies included, twenty-six were deemed to be of high quality, and five were of low quality and therefore at high risk of bias (Table 2).

### 3.1. AIP and Anthropometric Measurements

It is acknowledged that the only anthropometric measurement included as MetS diagnostic criteria by IDF is WC; however, we decided to still report other findings such as BMI and WHpR. 

WC was measured in 18 groups within seven different studies (Table 3). It was possible to establish that a high WC was found in five groups taken from four studies of good quality, and it was always associated with a high AIP. Further, this result was found in obese people, post-menopausal women, and apparently healthy females. It must be noted that the study that found high WC and high AIP in the latter group had participants with a healthy BMI in a student population, which questions the reliability of such results. On the other hand, more studies reported a WC that could have been deemed “high” if it was produced by data pertaining to women, but not men, and vice versa, depending on the ethnicity of participants (see Table 3). Therefore, this data could not be used to verify whether it was paired with a high AIP, as there was a risk it may not reflect true measurements. High WC was never found in groups with a medium or low AIP. The IDF WC criteria for MetS vary depending on ethnicity, which was largely unreported in the studies used, leading to the use of the study provenance instead to establish if thresholds were met.

In terms of BMI, shown in Table 4, twenty studies calculated this, five of which studied six groups of obese individuals. All of them also reported a high AIP, meaning that none of the obese groups had either a medium or a low AIP. Importantly, four out of five of these studies were of good quality. Furthermore, 17 groups of overweight people were studied by nine studies in total, eight of which were deemed to be of good quality. When divided based on AIP, the ones with a high AIP were pre- and post-menopausal women, patients with coronary artery disease (CAD), male smokers, T2DM patients, and only one healthy control group, and came from six studies of high quality and one of low quality. The overweight groups with a medium AIP were asymptomatic dyslipidaemic, male non-smokers, female smokers, normotensive, and hypertensive. However, 50% of the studies that produced this data were of low quality. Those with a low AIP (six groups) were all apparently healthy except for one group of people with one or two CMR factors, which would still not class them as MetS patients under any set of criteria exposed in Table 1. These results came from four studies out of five being of high quality. Subjects with a healthy BMI but a high AIP were composed of six healthy control groups, two CAD groups, and one hypertensive group. One of the studies that found this result in a healthy group was of low quality. Those with a medium AIP were two healthy control groups and one with newly diagnosed T2DM patients, taken from three good-quality studies. Finally, those with a low AIP comprised eight healthy/control groups, and three hypertensive groups, coming from five good-quality and two low-quality studies. 

WHpR was measured in eleven groups within five different studies. In one group, it was not possible to determine whether WHpR was high due to the same lack of gender stratification mentioned above. WHpR was found to be high in five groups from four studies, three of which were of good quality. In four groups out of five, WHpR was found together with a high AIP. The other study showed a medium AIP. No study that found a high WHpR reported a low AIP.

### 3.2. AIP and Blood Lipid Profile

It might appear of little use to verify if high TG and low HDL-C are frequently reported alongside a high AIP, since AIP = log^10^(TG/HDL-C). However, the logarithmic ratio itself does not provide information on whether the TG and HDL-C exceeded the thresholds for MetS diagnosis discussed previously. Thus, it was concluded that presenting these results would be of value. 

Blood lipids (Table 4) were measured in 72 groups between 25 studies. In total, 27 groups had high TG, 21 of which also had high AIP between 12 different studies. Interestingly, most of these groups were made up of people with diabetes, CAD, hypertension, and obesity. High TG and medium AIP were found in four groups, each from a different study, three of which were of good quality. These had their groups composed of patients with T2DM, dyslipidaemia, and hypertension. Notably, only the poor-quality study obtained this result in a healthy group. High TG and low AIP were found in two groups from two studies, one of which reported very large SD for both TG (2.24 ± 2.79 mmol/L) and AIP (0.05 ± 0.40).

Low HDL-C was found in 17 groups. Of them, nine also had high AIP and came from eight different studies. Out of nine, four groups were diabetics, two obese, one smokers, one post-menopausal women, and only one healthy group. No study found any group to have a low HDL-C and medium AIP. The remaining eight groups had low HDL-C and low AIP, and they all belonged to the same study. Unfortunately, most studies that measured HDL-C, in general, lacked gender stratification, making it therefore often challenging to establish if HDL-C could be considered “low” in many groups, as explained previously.

### 3.3. AIP and Blood Glucose

The majority of the studies screened stated that participants were taking glucose-lowering medication, or failed to confirm that they were not, and were therefore excluded from the current study. This meant that the data on AIP and IR were not as abundant as in previous examples. In total, seven studies measured IR-related parameters in 19 different groups (Table 5). Such data were quite heterogeneous in terms of methods and units used. Only one study measured HbA1c, five measured the % of participants affected by T2DM, one measured blood glucose and insulin, and only one measured insulin.

The study that measured HbA1c saw both its groups have the same HbA1c of 5.1%, which is not indicative of diabetes, although AIP was high in both groups, with the obese group having an AIP value of more than double that of the healthy weight control group [33]. 

Within each study, the higher the AIP, the higher the proportion of people with T2DM. For example, Guzel et al. (2021) [37] found a high AIP in both groups; however, the group with 43.3% diabetics had an AIP of 0.63 ± 0.25, while the group with 25.1% diabetics had an AIP of 0.48 ± 0.25. Altogether, seven groups had a high AIP, and in such groups, the proportion of diabetics varied between 11.72% and 45.3%. Three groups had a low AIP, and the proportion of diabetics varied between 4.9 and 9.1%. The one group with a medium AIP had 10.1% diabetics.

The two studies that measured insulin had a high AIP in all four groups, and in all of them, the higher the AIP, the higher the insulin.

### 3.4. AIP and Hypertension

Very few studies explicitly stated that their participants were not treated with BP-lowering medication, or that they were not on any medication, therefore greatly reducing the availability of BP-related measurements for synthesis. Only six studies were eligible, all of which were good quality, for a total of 18 groups (Table 6). Half of the studies measured SBP and DBP in ten groups, and the other half calculated the prevalence of hypertensive participants in the remaining eight groups. 

Of those measuring BP, one study found one group out of three to have high SBP. The group also had high AIP and was made of participants with either three or four CMR factors. One study had all its groups with raised SBP and DBP, but only one group had high AIP. Another study had the opposite situation, where no group was hypertensive on average; however, they all had raised AIP. Nevertheless, the latter two groups showed increasing BP with increasing AIP.

Of the three studies measuring hypertension prevalence, one found the CAD group to have high AIP and 63.5% hypertensive subjects, and the non-CAD group to have medium AIP and 43.2% hypertensive subjects. One had all its four groups with raised AIP, with the non-CAD group being 17.5% hypertensive, and the following three groups, in increasing order of CAD risk, to have 55.6%, 57.8%, and 56.8% hypertensive subjects, respectively. Finally, one found its group of T2DM and CAD subjects being 64.3% hypertensive and to have a high AIP, and its T2DM without CAD subjects to be 61.4% hypertensive and to have a low AIP.

## 4. Discussion

### 4.1. Interpretation of Main Findings and Pathophysiological Perspectives

Our findings reveal that AIP may be associated with WC, BMI, WHpR, TG, HDL-C, and IR; however, the results were less clear for HPT. 

All groups with a WC exceeding the IDF threshold for MetS had a high AIP, meaning that groups which had a medium or low AIP never had a concerning WC, which suggests a positive association between the two. BMI and WHpR were included in the results, despite not forming part of the IDF criteria for MetS, to provide context. The same result was seen in obese groups, with none having a low or medium AIP. Even groups that had a medium risk BMI, indicating overweight, often had a high AIP when participants had some characteristics which were relatable to MetS. However, overweight groups that lacked such characteristics tended to be slightly healthier overall. This trend seemed to translate across the BMI range, with healthy weight groups being healthier when the AIP was low. It might be argued, however, that BMI may not be the best indicator of excess weight, as it does not provide information on body composition, therefore making it impossible to distinguish between excess adipose tissue and muscle mass. Some studies have indicated that BMI is inferior to WC and WHpR for predicting CV events [56]. In our study, WHpR seemed to agree with other anthropometric findings, with four groups out of five presenting high WHpR in concordance with high AIP, and the final group having medium AIP. WC and WHpR provide an indication of how much visceral fat is likely to surround the abdomen. This has been identified as being more relevant than subcutaneous fat in the pathophysiology of MetS, and therefore, WC was deemed to be the best anthropometric measurement in the prediction of MetS [57]. Furthermore, AIP was found to be significantly (*p*-value < 0.001) associated with WC in many studies [58].

Almost all groups with high TG also had high AIP. This result was expected due to the mathematical function of TG in AIP; however, the TG being high enough to meet the MetS criteria could not have been predicted, making it therefore still an important result. It also corroborates the use of AIP to detect MetS, as TG concentration was found to be strongly associated with MetS components. The largest marginal correlation was found to be with BMI, mostly in women, and low HDL-C, mostly in men, with other correlated outcomes, also including BP [59]. Indeed, the current study found many groups with low HDL-C to also have a high AIP. This result was found in nine groups out of seventeen, with the remaining groups having low HDL-C together with low AIP. It must be stated that the latter groups, which differed from each other in terms of being shared equally across all BMI ranges, all came from the same study, which also had some other lipid measurements that may not be expected in those populations, thereby questioning the reliability of the findings. Therefore, it may be stated that HDL-C seems to be inversely associated with AIP. It was not surprising for low HDL-C to be found in groups with some aspects linked to MetS, as all subpopulations of this type of cholesterol display anti-oxidative, anti-inflammatory, anti-apoptotic, vasodilatory, anti-thrombotic, and anti-infectious actions [60], and low HDL-C was previously reported to be associated with MetS [61]. Most importantly, HDL-C plays an important role in the protective reverse cholesterol transport (RCT) process, which assist with the recycling or disposal of excess cholesterol, which also implicates TG. The RCT process has been described by several studies [60,62,63,64,65,66,67,68]. However, issues occur when TG is raised, as this increases the recently described rate of the lipid turnover [66]. Interestingly, in our study, HDL-C was found to be directly associated with cholesterol efflux, and inversely associated with WC [66]. This might provide an explanation for the high AIP seen in the current study groups with high WC. Moreover, obesity causes adipocytes’ ability to store TG as fat to increase, leading to fat being accumulated in the liver and muscles, causing issues such as non-alcoholic fatty liver disease (NAFLD) and IR [63].

The association between high WC and insulin resistance is well established [69,70]. This is likely due to the metabolic activity of white adipose tissue (WAT). It is common knowledge that the more WAT, the more leptin is produced, causing peripheral hyperleptinemia and central hypoleptinemia, resulting in hyperglycaemia, hyperinsulinemia, hyperlipidaemia, and inflammation. Visceral obesity is also related to decreased adiponectin, causing increased hepatic gluconeogenesis, IR, decreased skeletal muscle glucose uptake, and increased inflammation. In addition, obese individuals often experience decreased retinol-binding protein 4 and increased glucagon-like peptide 1, which again contribute to IR. Regardless of adiposity, low HDL-C was found to be an independent predictor of IR [71,72], and the same was found for TG [72,73]. In line with this, the current study found that AIP may be associated with IR. Moreover, all studies that calculated the percentage of diabetic subjects in their groups seem to show a trend of concurrent growth of this percentage and AIP, which is particularly interesting and should be assessed quantitatively to uncover any linear relationship. A similar pattern was seen with serum insulin, strengthening the findings. However, these may be interpreted with caution, as usually, when diabetes is present, patients also suffer from a series of cardiovascular comorbidities and are often overweight or obese, which make it difficult to determine if the association is indeed driven by IR or other CMR factors. The only study included here that measured HbA1c did not indicate diabetes for either of its two groups, both of which had high AIP. Interestingly, in the obese group, AIP was found to be more than twice as high as the control group, suggesting that perhaps the association is clearer for excess weight, although there could be many more reasons for such results, such as confounding factors or how measurements were performed [33]. Nevertheless, our study findings concerning AIP and IR are very encouraging and are backed by several other studies which found similar quantitative conclusions [20,74,75,76], with a very recent study highlighting how AIP is predictive of IR and recommending that AIP be used in clinical practice [77]. It is worth noting, however, that the same results were consistently not found in African Americans [78,79,80]. Due to a lack of comment regarding ethnicity by most studies included in our review, it is not possible to comment upon this; however, the authors acknowledge that it may be a potential confounding factor.

In terms of AIP and HPT, the results were less clear. All three studies included here that measured SBP and DBP produced very inconclusive results and did not suggest an association between AIP and BP. However, there may be many reasons for this. Importantly, BP is not a biomarker such as HbA1c, which provides a reliable overview of the patient’s blood glucose over several weeks. BP is physically measured by healthcare professionals who may have varying levels of expertise in taking the measurements. Kallioinen et al. (2016) [81] identified dozens more potential sources of error in BP measurements, including acute ingestion of food, alcohol, caffeine, or nicotine, bladder distension, cold exposure, device inaccuracy, body and arm position, and clothing effect. It is also possible that patients may feel anxious about their visit, especially if they are aware that their data are being collected for scientific research purposes, which might increase their heartbeat and BP. Conversely, the three studies included in our review, which calculated the percentage of hypertensive participants and AIP in their groups, yielded more interesting results. For example, there were 55.6–63.5% hypertensive subjects with high AIP in all CAD groups (five groups from three studies). The subjects in non-CAD groups had a medium AIP and 43.2% of them were hypertensive, and in those who had raised AIP, 17.5% were hypertensive. The other non-CAD group had low AIP and T2DM, and 61.4% were hypertensive. Although these groups were from a very limited number of studies and were relatively heterogeneous, AIP might be associated more with CAD itself than with hypertension. This would not be surprising, since AIP receives the most interest concerning CAD and CHD and has been significantly associated in many studies [20,43,82], and this relationship has also been confirmed by a recent meta-analysis [83]. Nevertheless, as previously explained, high AIP is driven by high TG and low HDL-C, which provide ideal conditions for the development of atherosclerosis, creating a milieu which is frequently accompanied by concomitant hypertension [84]. Furthermore, we also found high TG and low HDL-C to be positively associated with hypertension [85], as was AIP [15]. Therefore, the association between AIP and HPT should be investigated further to clarify the discrepancy between the current study’s findings and with the existing literature. 

### 4.2. Diet and Metabolic Syndrome

Our findings point towards an association between AIP and most CMR factors. This is particularly important given that most of the CMR are strongly modifiable by dietary and lifestyle factors; therefore, we deemed it necessary to highlight here the current knowledge on diet and MetS. As summarised below, it cannot be concluded with certainty if AIP predicts MetS; however, the nutritional advice that practitioners could give to individuals with high AIP and no apparent MetS may still be helpful, as it mostly reflects healthy eating advice recommended by many international organisations to all individuals. Such advice would also be beneficial to individuals who are already displaying some traits of the MetS, as dietary and lifestyle modifications were shown to facilitate disease regression [86].

It is undeniable that given the central role of WC in the development of MetS, any dietary advice given should first and foremost focus on weight reduction if the patient has a BMI > 25 kg/m^2^, as they would be classed overweight or obese [86,87,88,89,90]. This can be achieved in many ways, and typically involves calorie restriction, increased calorie expenditure, and changes in terms of diet composition. Rochlani et al. (2017) [89] recommended a weight loss of 7–10% in baseline body weight over a 6–12 month time frame, which may be achieved through a reduction of caloric intake by 500–1000 kcal/day, with another study [90] also recommending an energy deficit of around 500 kcal/day for this population. Some studies also focused on eating patterns and the frequency of MetS. For example, Ha et al. (2019) [91] reported a reduced incidence of MetS in participants who ate in the morning (OR = 0.73 in men and 0.69 in women) compared to those who did not, and the opposite result was found in men who ate at night (OR = 1.48), with very similar results found in women (OR = 1.68) in a different study [92]. There seems to be an agreement around the notion that small and frequent meals are preferred over fewer but larger ones due to improvements seen in both blood glucose, lipid profile, and obesity, which was proposed to possibly be due to reduced glucose and insulin spikes [90,93]. Furthermore, it has also been found that [94] the higher the number of CMR factors, the lower the incidence of the regular eating pattern (OR = 0.27) and the higher the IR (OR = 0.68). A recent review of long cohort studies [93] found that participants were more likely to develop MetS later in life if they had an irregular meal pattern when adolescent, which is interesting from a prevention perspective and is highly relevant to the educational role of nutrition professionals. The same authors also reviewed intermitted fasting for MetS and found that the weight loss was significant (*p* < 0.001), although changes in blood glucose and blood lipid profile were not present.

It is generally accepted that the most appropriate diet for MetS closely reflects the main features of the Mediterranean diet [95]. These include the presence of considerable amounts of olive oil, as well as wheat, grapes, and their derivative products, complex carbohydrates, varied types of fibre, and quite a high proportion of energy coming from fat (40%), which is mostly unsaturated [88]. The Mediterranean diet is also low in processed food, and therefore in salt, which is very relevant to the pathophysiology of hypertension. Studies found a 14–25% reduction in the odds of developing MetS when following this dietary pattern [86,88,95], and a meta-analysis showed a 49% increased probability of remission from the metabolic syndrome [96]. These results are thought to be driven by the antioxidant and anti-inflammatory properties of foods which characterise the Mediterranean diet, such as food items which are high in fibre [88]. 

Many authors attempted to investigate the relationship of MetS with macro and micronutrients, and even some non-nutrient compounds. An umbrella review of 30 systematic reviews and meta-analyses found highly suggestive evidence that carbohydrate intake is associated with a high risk of developing MetS [97]. Unfortunately, this study failed to distinguish between different carbohydrate types, which is very relevant when evidence is translated into practical dietary advice. A systematic review and meta-analysis found sugar-sweetened beverages to be associated with MetS, although the authors warn that an unhealthy lifestyle might be a major confounding factor [98]. On the other hand, complex carbohydrates may be beneficial. Interestingly, two meta-analyses on MetS and fibre published around the same time came to different conclusions. One of them [99] found an inverse relationship between MetS risk and dietary fibre consumed, while the other [100] reached a more cautious conclusion, warning that the available evidence is not sufficient to be certain about the protective effect of fibre on MetS risk. However, they both present some physiological explanations in favour of fibre. They explain that it may have a positive effect on obesity due to a decreased energy density of food rich in fibre, which may also suppress appetite and delay gastric emptying. Viscous soluble fibre may lower cholesterol through faecal bile salts excretion and delayed and reduced absorption; however, they warn that the effect of fibre on reducing TG or increasing HDL-C is not conclusive. Overall, cereal and fruit fibre seem to be more beneficial than vegetable fibre for MetS, but intakes much higher than 30 g/day do not appear to provide any additional benefits [99]. 

In terms of fat, overall recommendations have moved away from a low-fat diet to prevent or treat MetS, as the energy deducted from the decreased fat consumption must be replaced by something else, which is usually an excessive intake of dietary carbohydrates. The attention is now far more focused on fat quality. Saturated fat, present in large amounts in animal products, should be limited to <10% daily energy, as it promotes dyslipidaemia and therefore atherosclerosis, and the remaining 15–25% should derive from mono and polyunsaturated (PUFA) sources found in vegetable oils, fish, nuts, and seeds [88]. Omega 3 fatty acids have also been shown to have a positive impact on both triglycerides and HDL-C [90].

Protein received less interest in relation to MetS. It is advised to consume around 15% of total daily energy from protein. It is widely recognised now that high-protein diets are not harmful to kidney health as once thought; however, high-protein diets should be avoided in the MetS population as there is susceptibility to kidney disease [90].

### 4.3. Strengths and Limitations

The current review has some considerable strengths. Most importantly, it involved the support of a second reviewer whenever there was doubt about the inclusion or exclusion of certain studies, therefore reducing selection bias and the chances of errors being made. Moreover, strict inclusion and exclusion criteria were set, which ensured that many confounding factors would be avoided, and the results would reflect the population of interest as homogeneously as possible. Furthermore, it might be stated that the number of healthy and/or control groups and that of groups with some aspect of MetS were proportionate, which ensured a range of values as broad as possible, making it easier to determine if an association between two variables would or would not be suspected. Finally, the study produced some very interesting and robust results that can be a valuable starting point for future research.

Conversely, our review also has some limitations which should be acknowledged. Perhaps the most important is that only observational studies were included, which were either cross-sectional or case–control studies. This made it impossible to determine any cause–effect relationships. This has major repercussions on any practice recommendations. Indeed, it would be unjustified to recommend using AIP as a detection method for MetS, as AIP may increase as a result of a growing WC or IR, with a delay that cannot be known. Another limitation is the lack of comments regarding the ethnicity of participants in most studies, many of which were based in Asian countries. If the assumption is that their population was also mainly Asian, then this might have affected the generalizability and external validity of the results produced here. Most studies also did not stratify their results based on gender, which made it challenging to establish if some MetS criteria thresholds were crossed, as this might have been the case if the data came from men but not if it came from women and vice versa. This might have led to the under-reporting of CMR–AIP associations. The current review focused on the AIP and its association with cardiometabolic risk; however, it is important to consider that AIP is only one of the potential indices in this regard, and future studies can consider the diagnostic power of other markers, such as non-HDL-C in comparison and/or in combination with AIP [101]. Finally, although efforts were made to limit confounding factors, it is acknowledged that blood lipid results might have been impacted by participants being smokers [102], which was not an exclusion criterion.

### 4.4. Implications for Practice and Direction of Future Research

The current review included only observational studies. Therefore, its results could not have the level of certainty that would be required for AIP to be recommended in clinical practice yet. The current study produced interesting evidence; however, correlations could not be quantified due to the qualitative nature of the review. Thus, it is suggested that the current study results are used to inform future meta-analyses. Alternatively, it would also be very useful to have more cohort studies to ascertain the cause–effect of AIP and MetS. In this case, it is recommended that the study is stratified based on gender to simplify the detection of values that meet the MetS threshold and make full use of the available data. In addition, it is generally agreed that IR has a major role in MetS. Therefore, due to the lack of universal agreement on the association between AIP and IR in different ethnicities, future studies should also consider the role of this and stratify their results by ethnicity.

## 5. Conclusions

Our review aimed to qualitatively summarise observational data on AIP and CMR factors and to verify their association, making this the first study of this kind. After performing a systematic search of PubMed, One Search, and the Cochrane library, we included 32 quality-assessed studies. These studies revealed a possible association between AIP and WC, TG, HDL-C, and IR. However, no association was observed between AIP and HPT. The results of the current study should be used to inform future meta-analyses, which will offer further insights into the associations. Furthermore, additional cohort studies stratified by gender and ethnicity are recommended, together with the recruitment of newly diagnosed HPT patients who are not yet on medication.

## Figures and Tables

**Figure 1 healthcare-11-00966-f001:**
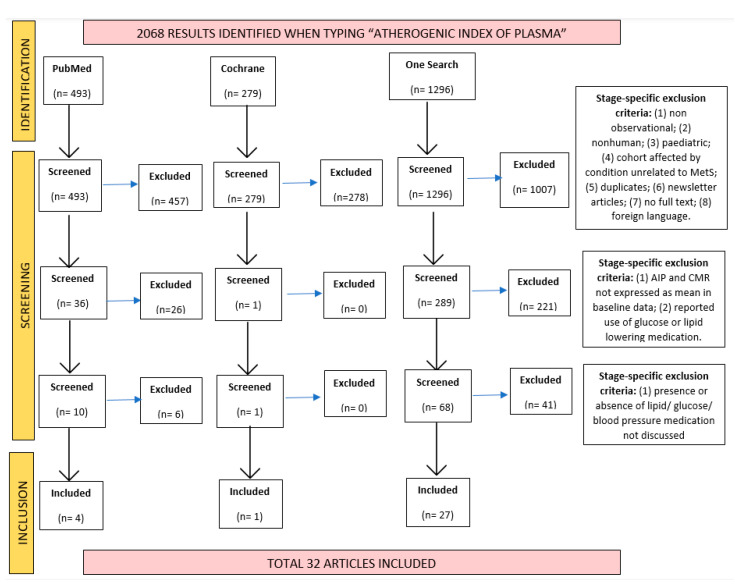
PRISMA flowchart showing the studies identified, screened and included in the current systematic review, and the exclusion criteria applied at each of these stages.

**Table 2 healthcare-11-00966-t002:** Quality assessment of all 30 studies included using the Newcastle-Ottawa quality assessment scales for case–control studies and cross-sectional studies (adapted) [27,28,29,30,31,32,33,34,35,36,37,38,39,40,41,42,43,44,45,46,47,48,49,50,51,52,53,54,55].

	Newcastle-Ottawa Quality Assessment Criteria				
Author and Year	Selection (MAX *** or **** Depending on Study Design)	Comparability (MAX **)	Outcome (MAX **)	Exposure (MAX ***)	Total Score
**Eslami et al., 2019**	**/3	**	**	/	6/7
**Ranjit et al., 2015**	***/4	**	/	**	7/9
**ChhodenR et al., 2021**	*/3	**	**	/	5/7
**Chhezom et al., 2017**	*/3	*	**	/	4/7
**Panjeta et al., 2018**	*/3	*	**	/	4/7
**Bahijri et al., 2020**	*/3	**	**	/	5/7
**Hanamatsu et al., 2014**	***/4	**	/	**	7/9
**Olamoyegun et al., 2016**	*/3	*	**	/	4/7
**Manohar et al., 2013**	***/4	**	/	**	7/9
**Agrawall et al., 2014**	*/3	**	**	/	5/7
**Guzel et al., 2021**	***/4	**	/	***	8/9
**Krivosikova et al., 2015**	*/3	**	**	/	5/7
**Li et al., 2015**	**/3	**	**	/	6/7
**Cibičková et al., 2019**	**/3	**	*	/	5/7
**Vaverkova et al., 2017**	**/3	**	**	/	6/7
**Al-Bazi et al., 2011**	/3	*	**	/	3/7
**Wang et al., 2020**	**/3	**	**	/	6/7
**Yin et al., 2021**	***/3	**	**	/	7/7
**Choudhary et al., 2019**	**/3	**	**	/	6/7
**Javardi et al., 2020**	**/3	*	**	/	5/7
**Mondal et al., 2021**	**/3	**	**	/	6/7
**Anandkumar et al., 2019**	*/3	*	*	/	3/7
**Nam et al., 2021**	**/3	**	**	/	6/7
**Hosseini et al., 2020**	**/3	**	**	/	6/7
**Cakirca and Celik, 2019**	*/3	**	**	/	5/7
**Lwow and Bohdanowicz-Pawlak, 2020**	**/3	**	**	/	6/7
**Wang et al., 2016**	*/3	**	**	/	5/7
**Al-Shaer et al., 2021**	*/3	**	**	/	5/7
**Nwagha et al., 2010**	****/4	**	/	*	7/9

A study can be awarded a maximum of one star for each numbered item within the Selection, Exposure and Outcome categories. A maximum of two stars can be given for Comparability.

**Table 3 healthcare-11-00966-t003:** Review of studies interlinking anthropometric parameters and AIP. Author, year, subject characteristics, study design, primary outcomes, and other study outcomes of interest (BMI, WC, WHpR, WHtR) are presented for each study where available [27,28,29,30,33,35,38,39,41,42,43,44,45,46,47,48,49,50,52,53].

Author and Year	Population Characteristics and Ethnicity/Provenance	Design	Study Outcomes
**Eslami et al., 2019**	310 apparently healthy undergraduate students; 18–25 years old; *Iran*	Cross-sectional study	Males: BMI (kg/m^2^): 22.90 ± 3.53 WC (cm): 86.37 ± 8.60 WHpR: 0.88 ± 0.05 (*high?*) WHtR: 0.50 ± 0.05 AIP: 0.34 ± 0.16 (high) Females: BMI (kg/m^2^): 23.50 ± 3.47 WC (cm): 80.74 ± 7.34 (*high*) WHpR: 0.81 ± 0.04 WHtR: 0.50 ± 0.04 AIP: 0.26 ± 0.17 (*high*)
**Ranjit et al., 2015**	75 women; Pre-menopausal women with normal lipid profile (*n* = 51; mean age 39.72 ± 1.53 years) Post-menopausal women with elevated lipid profile (*n*= 24; mean age 62.21 ± 1.46) *India*	Not stated	Pre-menopausal women: BMI (kg/m^2^): 25.06 ± 0.65 (*medium*) WC (cm): 34.68 ± 0.60 HC (cm): 52.43 ± 1.56 WHpR: 0.68 ± 0.01 AIP: 0.40 ± 0.02 (*high*) Post-menopausal women: BMI (kg/m^2^): 27.98 ± 1.58 (*medium*) WC (cm): 42.63 ± 0.53 HC (cm): 68.25 ± 1.50 WHpR: 0.63 ± 0.01 AIP: 0.52 ± 0.04 (*high*)
**ChhodenR et al., 2021**	64 adults suspected of coronary atherosclerosis; CAD group (*n* = 42; mean age 49.3 ± 8.0 years; 38 males/4 females) had at least one stenosis (≥20%) Control group (*n* = 22; mean age 49.5 ± 8.6; 10 males/12 females) had <20% stenosis *Bangladesh*	Cross-sectional study	CAD group: BMI (kg/m^2^): 24.77 ± 2.8 AIP: 0.43 (*high*) Control group: BMI (kg/m^2^): 26.08 ± 3.2 (*medium*) AIP: 0.19 (*medium*)
**Chhezom et al., 2017**	90 apparently healthy subjects; Normal weight (*n* = 22; mean age 40.86 ± 10.26 years; 11 males/11 females) Overweight (*n* = 24; mean age 43.67 ± 9.87 years; 13 males/11 females) Obese (*n* = 44; mean age 41.43 ± 9.84 years; 21 males/23 females) *Bangladesh*	Cross-sectional study	Normal weight: BMI (kg/m^2^): 21.21 ± 1.15 AIP: 0.053 Overweight: BMI (kg/m^2^): 25.46 ± 1.38 (*medium*) AIP: 0.281 (*high*) Obese: BMI (kg/m^2^): 31.10 ± 2.28 (*high*) AIP: 0.331 (*high*)
**Hanamatsu et al., 2014**	23 students of 18–27 years; Obese (*n* = 12, mean age 21.7 ± 2.8 years, 12 females/0 males) Healthy weight (*n* = 11, mean age 23.2 ± 2.4, 7 males/4 females) *Japan*	Not stated	Obese: BMI (kg/m^2^): 36.8 ± 4.4 (*high*) AIP: 0.81 ± 0.28 ** (*high*) Healthy weight control: BMI (kg/m^2^): 19.8 ± 1.1 AIP: 0.30 ± 0.26 ** (*high*)
**Manohar et al., 2013**	104 subjects; Newly diagnosed T2DM (*n* = 52, mean age 50.2 ± 1.13 years) Controls (*n* = 53, mean age 52.3 ± 0.95 years) *India*	Not stated	Newly diagnosed T2DM: BMI (kg/m^2^): 24.87 ± 0.70 AIP: 0.16 ± 0.03 * (*medium*) Controls: BMI (kg/m^2^): 23.92 ± 0.47 AIP: 0.08 ± 0.01 * (*low*)
**Krivosikova et al., 2015**	411 apparently healthy adults; 0 CMR factors (*n* = 162, mean age 30.0 ± 8.9 *** years, 20% males) 1–2 CMR factors (*n* = 162, mean age 36.2 ± 13.9 years, 46% males) 3–4 CMR factors (*n* = 87, mean age 46.1 ± 14.6, 72% males) *Slovakia* Note: *p*-value compared to 0 CMR factors group. Not possible to establish if WC = high due to lack of gender stratification.	Cross-sectional study	0 CMR factors: BMI (kg/m^2^): 21.6 ± 2.1 WC (cm): 75.0 ± 6.7 AIP: −0.30 ± 0.21 (*low*) 1–2 CMR factors: BMI (kg/m^2^): 25.7 ± 4.1 *** (*medium*) WC (cm): 87.9 ± 11.3 *** (*high?*) AIP: −0.10 ± 0.30 *** (*low*) 3–4 CMR factors: BMI (kg/m^2^): 30.2 ± 4.3 *** (*high*) WC (cm): 104.1 ± 11.0 *** (*high*) AIP: 0.28 ± 0.24 *** (*high*)
**Li et al., 2015**	1772 clinically suspected CAD cases; CAD diagnosed (≥50% coronary lesion, *n* = 1057, mean age 58.03 ± 10.0 years, 68.4% males) CAD not diagnosed (*n* = 715, mean age 52.56 ± 11.67 years, 49.5% males) *China*	Cross-sectional study	CAD diagnosed: BMI (kg/m^2^): 25.61 ± 3.35 *** (*medium*) AIP: 0.23 ± 0.26 *** (*high*) CAD not diagnosed: BMI (kg/m^2^): 24.87 ± 3.44 *** AIP: 0.13 ± 0.33 *** (*medium*)
**Vaverkova et al., 2017**	607 asymptomatic dyslipidemic patients (mean age 45.6 ± 14.0 years, 295 males/312 females) *Czech Republic* Note: Not possible to establish if WC was high due to lack of gender stratification.	Not stated	BMI (kg/m^2^): 26.3 ± 4.0 (*medium*) WC (cm): 88.3 ± 12.6 (*high*?) AIP: 0.13 (*medium*)
**Al-Bazi, 2011**	106 healthy adults age range 21–45 years; Female non-smokers (mean age 36.3 ± 12.1 years) Male non-smokers (mean age 34.3 ± 12.7 years) Female smokers (mean age 34.6 ± 7.5 years) Male smokers (mean age 35.1 ± 12.7) *Saudi Arabia* Note: Total non-smokers *n* = 51; total smokers *n*= 55.	Not stated	Female non-smokers: BMI (kg/m^2^): 24.2 ± 3 AIP: 0.09 ± 0.05 Male non-smokers: BMI (kg/m^2^): 25.9 ± 2.0 (*medium*) AIP: 0.13 ± 0.09 (*medium*) Female smokers: BMI (kg/m^2^): 25.5 ± 3.3 (*medium*) AIP: 0.35 ± 0.13 (*medium*) Male smokers: BMI (kg/m^2^): 27 ± 4.8 (*medium*) AIP: 0.34 ± 0.11 (*high*)
**Wang et al., 2020**	3600 suspected CAD cases divided based on SYNTAX score; Non-CAD (*n* = 1347, mean age 58.7 ± 8.9 years, 62.3% males) 1 ≤ SS ≤ 23 (*n* = 1448, mean age 61.7 ± 10.7 years, 61.3% males) 23 ≤ SS ≤ 33 (*n* = 569, mean age 61.2 ± 10.4 years, 61.9% males) SS ≥ 33 (*n* = 236, mean age 60.5 ± 11.4 years, 69.9% males) *China* Note: *p*-value compared to non-CAD group.	Not stated	Non-CAD BMI (kg/m^2^): 24 ± 3.5 AIP: 1.9 ± 0.2 (*high*) 1 ≤ SS ≤ 23 BMI (kg/m^2^): 24.3 ± 3.4 ** AIP: 2.1 ± 0.2 (*high*) 23 ≤ SS ≤ 33 BMI (kg/m^2^): 24.8 ± 3.2 ** AIP: 2.2 ± 0.2 (*high*) SS ≥ 33 BMI (kg/m^2^): 25.9 ± 3.7 ** (*medium*) AIP: 2.3 ± 0.3 (*high*)
**Yin et al., 2021**	4744 Chinese individuals with hypertension stratified based on AIP; Q1 (*n* = 1184, mean age 66.89 ± 9.11 years, 60.47% males) Q2 (*n* = 1187, mean age 65.67 ± 9.16 years, 48.53% males) Q3 (*n* = 1187, mean age 63.65 ± 9.17 years, 42.63% males) Q4 (*n* = 1186, mean age 61.68 ± 9.32 years, 47.89% males) *China* Note: Not possible to establish if WC = high due to lack of gender stratification.	Cross-sectional study	Q1: BMI (kg/m^2^): 21.38 ± 3.09 *** WC (cm): 76.46 ± 8.89 *** AIP: −0.00 ± 0.28 *** (*low*) Q2: BMI (kg/m^2^): 22.72 ± 3.31 *** WC (cm): 80.76 ± 9.18 *** (high?) AIP: −0.11 ± 0.05 *** (*low*) Q3: BMI (kg/m^2^): 24.10 ± 3.40 *** WC (cm): 84.49 ± 8.81 *** (*high*?) AIP: 0.08 ± 0.06 *** (low) Q4: BMI (kg/m^2^): 24.81 ± 3.08*** WC (cm): 86.80 ± 7.87 *** (*high?*) AIP: 0.37 ± 0.14 *** (*high*)
**Choudhary et al., 2019**	615 normotensive (40.5%) AND never-treated subjects with primary hypertension 59.5%); T1 (*n* = 202, mean age 44.7 ± 12.3 years, 104 males/98 females) T2 (*n* = 208, mean age 44.1 ± 11.9 years, 106 males/102 females) T3 (*n* = 205, mean age 44.9 ± 11.6, 104 males/101 females) *Finland* Note: *p*-values compared to T1.	Cross-sectional study	T1: BMI (kg/m^2^): 25.1 ± 3.7 (*medium*) AIP: −0.44 ± 0.17 (*low*) T2: BMI (kg/m^2^): 26.4 ± 4.0 * (*medium*) AIP: −0.17 ± 0.16 * (*low*) T3: BMI (kg/m^2^): 28.9 ± 4.7 * (*medium*) AIP: 0.15 ± 0.23 * (*medium*)
**Javardi et al., 2020**	157 individuals, age range 18–65 years, divided by weight; Healthy weight (*n* = 71, mean age 38.90 ± 10.976 years, 80.3% males) Overweight and obese (*n* = 86, mean age 38.60 ± 9.394 years, 81.6% males) *Iran*	Cross-sectional, descriptive-analytic case–control study	Normal weight: BMI (kg/m^2^): 24.57 ± 2.32 WHpR: 0.90 ± 0.04 (*high*) AIP: 0.170 ± 0.07 (*medium*) Overweight and obese: BMI (kg/m^2^): 30.28 ± 3.16 *** (*high*) WHpR: 0.95 ± 0.06 *** (*high*) AIP: 0.214 ± 0.111 * (*high*)
**Mondal et al., 2021**	140 patients newly diagnosed with T2DM; 57.6% males; Over 70% primary/below primary education *India*	Cross-sectional study	BMI (kg/m^2^): 28.29 ± 2.47 (*medium*) AIP: 0.57 ± 0.07 (*high*)
**Anandkumar et al., 2019**	60 apparently healthy young females, age range 18–30 years; *India*	Cross-sectional study	BMI (kg/m^2^): 24.408 ± 3.078 WC (cm): 79.359 ± 4.65 WHpR: 0.85275 ± 0.03 (*high*) AIP: 0.581 ± 0.148 (*high*)
**Nam et al., 2021**	3468 healthy Koreans, stratified based on AIP; Q1 (*n* = 868, mean age 50.3 ± 9.6 years, 242 males/626 females) Q2 (*n* = 878, mean age 52.3 ± 9.1 years, 417 males/461 females) Q3 (*n* = 880, mean age 52.9 ± 8.6 years, 606 males/274 females) Q4 (*n* = 842, mean age 51.6 ± 8.8 years, 705 males/137 females) *Korea*	Cross-sectional study	Q1: BMI (kg/m^2^): 21.7 ± 2.6 AIP: −0.04 ± 0.11 (*low*) Q2: BMI (kg/m^2^): 23.0 ± 3.1 *** AIP: −0.02 ± 0.06 *** (*low*) Q3: BMI (kg/m^2^): 23.8 ± 2.7*** AIP: 0.02 ± 0.06 *** (*low*) Q4: BMI (kg/m^2^): 25.1 ± 2.7 *** (*medium*) AIP: 0.32 ± 0.13 *** (*high*)
**Hosseini et al., 2020**	183 women; age range 20–35 years; stratified based on weight and metabolic state; Metabolically healthy normal weight (*n* = 53, mean age 26.36 ± 5.01 years) Normal weight obese (*n* = 29, mean age 27.21 ± 4.61 years) Metabolically healthy obese (*n* = 57, mean age 28.44 ± 4.53 years) Metabolically unhealthy obese (*n* = 44, mean age 27.09 ± 4.30 years) *Iran* Note: Normal weight obese = BMI < 25 kg/m^2^ and body fat >30%. * = *p*-value between 1st and 3rd groups; ^ = *p*-value between 3rd and 4th groups.	Cross-sectional study	Metabolically healthy normal weight BMI (kg/m^2^): 22.14 ± 1.85 *** WC (cm): 71.20 ± 5.65 *** HC (cm): 93.04 ± 5.09 *** WHpR: 0.76 ± 0.06 *** AIP: 0.61 ± 0.19 *** (*high*) Normal weight obese BMI (kg/m^2^): 23.29 ± 1.21 WC (cm): 74.25 ± 4.53 HC (cm): 95.72 ± 4.48 WHpR: 0.77 ± 0.05 AIP: 0.63 ± 0.24 (*high*) Metabolically healthy obese BMI (kg/m^2^): 33.19 ± 6.37 *** (*high*) WC (cm): 95.82 ± 12.90 ***^^^ (*high*) HC (cm): 111.03 ± 9.11 ***^^^ WHpR: 0.86 ± 0.07 *** (*high*) AIP: 0.67 ± 0.17 *** (*high*) Metabolically unhealthy obese BMI (kg/m^2^): 37.17 ± 5.53 (*high*) WC (cm): 102.29 ± 9.79 ^^^ (*high*) HC (cm): 116.25 ± 7.40 ^^^ WHpR: 0.87 ± 0.05 (*high*) AIP: 0.96 ± 0.17 (*high*)
**Lwow and Bohdanowicz-Pawlak, 2020**	318 post-menopausal Polish women of Caucasian origin (mean age 55.3 ± 2.8 years) *Poland*	Not stated	BMI (kg/m^2^): 27.3 ± 4.7 (*medium*) WC (cm): 87.6 ± 11.6 (*high*) AIP: 0.35 ± 0.58 (*high*)
**Wang et al., 2016**	1475 adults stratified by gender; Men (*n* = 829, mean age 40 years ***) Women (*n* = 646, mean age 38 years ***) *China*	Not stated	Men: BMI (kg/m^2^): 25.69 (*medium*) AIP: 0.10 (*low*) Women: BMI (kg/m^2^): 22.38 *** AIP: −0.18 *** (*low*)

* *p* ≤ 0.05, ** *p* ≤ 0.01, *** *p* ≤ 0.001, ^ = the *p*-value between groups.

**Table 4 healthcare-11-00966-t004:** Outline of the studies interlinking blood lipid markers and AIP. Author, year, subject characteristics, study design, primary outcomes, and other study outcomes of interest (TC, TG, LDL-C, HDL-C) are presented for each study where available [31,32,33,34,35,36,37,38,39,40,41,42,43,44,45,46,47,48,49,50,51,52,53,54,55].

Author and Year	Population Characteristics and Ethnicity/Provenance	Design	Study Outcomes
**Panjeta et al., 2018**	60 adults with T2DM for ≥5 years; HbA1C ≤ 7% (*n* = 32; age range 38–94 years; 18 females/14 males) HbA1c ≥ 7% (*n* = 28; age range 32–87 years; 12 females/16 males) *Bosnia and Herzegovina* Note: Not possible to determine if HDL is low in the second group due to lack of gender stratification.	Cross-sectional	HbA1C ≤ 7% TC (mmol/L): 5.12 ± 1.20 (mg/dL): 197.99 ± 46.40 TG (mmol/L): 1.48 (range: 1.15–2.22); (mg/dL): 131.09 (range: 44.47–85.85) LDL-C (mmol/L): 3.10 ± 1.18 (mg/dL): 119.99 ± 45.63 HDL-C (mmol/L): 1.10 (range: 0.82–1.36) (mg/dL): 42.44 (range: 31.71–52.59) AIP: 1.75 ± 0.19 (*high*) HbA1c ≥ 7% TC (mmol/L): 5.34 ± 1.59 (mg/dL): 206.50 ± 61.10 TG (mmol/L): 2.15 (range: 1.40–3.32) (*high*) (mg/dL): 190.43 (range: 54.14–128.38) LDL-C (mmol/L): 3.20 ± 1.54 (mg/dL): 123.74 ± 59.55 HDL-C (mmol/L): 1.00 (range: 0.80–1.97) (low?) (mg/dL): 38.67 (range: 30.94–76.18) AIP: 2.79 ± 0.42 (*high*)
**Bahijri et al., 2020**	98 apparently healthy adults stratified by weight (although only mean BMI provided);Age range 18–55; Males (*n* = 48; mean age 27.81 ± 8.15 years) Females (*n* = 50; mean age 25.22 ± 9.2 years) *Saudi Arabia*	Cross-sectional study	Underweight males: TC (mg/dL): 79.75 ± 9.39 (mmol/L): 2.06 ± 0.24 TG (mg/dL): 22.33 ± 13.2 (mmol/L): 0.25 ± 0.15 LDL-C (mg/dL): 53.24 ± 8.30 (mmol/L): 1.38 ± 0.21 HDL-C (mg/dL): 22.05 ± 3.20 (*low*) (mmol/L): 0.57 ± 0.08 AIP: −0.33 ± 0.18 (*low*) Underweight females: TC (mg/dL): 69.17 ± 10.63 (mmol/L): 1.79 ± 0.27 TG (mg/dL): 10.12 ± 3.60 (mmol/L): 0.11 ± 0.04 LDL-C (mg/dL): 39.44 ± 8.00 (mmol/L): 1.02 ± 0.21 HDL-C (mg/dL): 27.69 ± 2.73 (*low*) (mmol/L): 0.72 ± 0.07 AIP: −0.45 ± 0.13 (*low*) Healthy weight males: TC (mg/dL): 79.99 ± 16.22 (mmol/L): 2.07 ± 0.42 TG (mg/dL): 15.35 ± 6.22 (mmol/L): 0.17 ± 0.07 LDL-C (mg/dL): 54.97 ± 13.83 (mmol/L): 1.42 ± 0.36 HDL-C (mg/dL): 21.94 ± 3.64 (*low*) (mmol/L): 0.57 ± 0.09 AIP: −0.32 ± 0.14 (*low*) Healthy weight females: TC (mg/dL): 76.13 ± 9.98 (mmol/L): 1.97 ± 0.26 TG (mg/dL): 13.23 ± 3.43 (mmol/L): 0.15 ± 0.03 LDL-C (mg/dL): 49.24 ± 9.41 (mmol/L): 1.27 ± 0.24 HDL-C (mg/dL): 24.23 ± 2.78 (*low*) (mmol/L): 0.63 ± 0.07 AIP: −0.27 ± 0.13 (*low*) Overweight males: TC (mg/dL): 89.37 ± 21.36 (mmol/L): 2.31 ± 0.55 TG (mg/dL): 18.23 ± 6.75 (mmol/L): 0.21 ± 0.08 LDL-C (mg/dL): 65.36 ± 20.18 (mmol/L): 1.68 ± 0.52 HDL-C (mg/dL): 20.36 ± 2.84 (*low*) (mmol/L): 0.53 ± 0.07 AIP: −0.36 ± 0.13 (*low*) Overweight females: TC (mg/dL): 83.38 ± 23.14 (mmol/L): 2.16 ± 0.60 TG (mg/dL): 12.39 ± 5.58 (mmol/L): (0.14 ± 0.06) LDL-C (mg/dL): 54.36 ± 18.90 (mmol/L): 1.41 ± 0.49 HDL-C (mg/dL): 26.54 ± 5.29 (*low*) (mmol/L): 0.69 ± 0.14 AIP: −0.36 ± 0.19 (*low*) Obese males: TC (mg/dL): 96.04 ± 17.58 (mmol/L): 2.33 ± 0.45 TG (mg/dL): 34.84 ± 20.34 (mmol/L): 0.39 ± 0.23 LDL-C (mg/dL): 68.99 ± 15.58 (mmol/L): 1.78 ± 0.40 HDL-C (mg/dL): 20.08 ± 5.01 (*low*) (mmol/L): 0.52 ± 0.13 AIP: −0.28 ± 0.25 (*low*) Obese females: TC (mg/dL): 88.47 ± 16.98 (mmol/L): 2.29 ± 0.44 TG (mg/dL): 15.12 ± 5.38 (mmol/L): 0.17 ± 0.06 LDL-C (mg/dL): 62.02 ± 14.86 (mmol/L): 1.60 ± 0.38 HDL-C (mg/dL): 23.42 ± 3.97 (*low*) (mmol/L): 0.61 ± 0.10 AIP: −0.21 ± 0.17 (*low*)
**Hanamatsu et al., 2014**	23 students of 18–27 years; Obese (*n* = 12, mean age 21.7 ± 2.8 years, 12 females/0 males) Healthy weight (*n* = 11, mean age 23.2 ± 2.4, 7 males/4 females) *Japan*	Not stated	Obese: TC (mg/dL): 204.8 ± 71.4 (mmol/L): 5.30 ± 1.85 TG (mg/dL): 130.1 ± 75.5 (mmol/L): 1.47 ± 0.85 LDL-C (mg/dL): 143.3 ± 60.4 * (mmol/L): 3.71 ± 1.56 HDL-C (mg/dL): 40.3 ± 5.2 ** (*low*) (mmol/L): 1.04 ± 0.13 AIP: 0.81 ± 0.28 ** (*high*) Healthy weight control: TC (mg/dL): 171.6 ± 19.9 (mmol/L): 4.44 ± 0.51 TG (mg/dL): 62.6 ± 38.3 (mmol/L): 0.71 ± 0.43 LDL-C (mg/dL): 90.4 ± 24.5 * (mmol/L): 2.34 ± 0.63 HDL-C (mg/dL): 64.6 ± 13.6 ** (mmol/L): 1.67 ± 0.35 AIP: 0.30 ± 0.26 ** (*high*)
**Olamoyegun et al., 2016**	106 acute stroke patients; Haemorrhagic stroke Ischemic stroke *Nigeria* Note: Not possible to determine if HDL = low in both groups due to lack of gender stratification.	Retrospective descriptive cross-sectional study	Haemorrhagic stroke: TC (mg/dL): 165.06 ± 36.66 (mmol/L): 4.28 ± 0.95 TG (mg/dL): 106.56 ± 44.76 (mmol/L): 1.20 ± 0.51 LDL-C (mg/dL): 104.18 ± 35.79 (mmol/L): 2.68 ± 0.93 HDL-C (mg/dL): 40.69 ± 13.76 (*low*?) (mmol/L): 1.05 ± 0.36 AIP: 0.62 ± 0.10 (*high*) Ischemic stroke: TC (mg/dL): 193.55 ± 71.91 (mmol/L): 5.01 ± 1.86 TG (mg/dL): 126.87 ± 73.04 (mmol/L): 1.43 ± 0.82 LDL-C (mg/dL): 129.74 ± 71.13 (mmol/L): 3.36 ± 1.84 HDL-C (mg/dL): 41.87 ± 15.33 (*low*?) (mmol/L): 1.08 ± 0.40 AIP: 0.68 ± 0.20 (*high*)
**Manohar et al., 2013**	104 subjects; Newly diagnosed T2DM (*n* = 52, mean age 50.2 ± 1.13 years) Controls (*n* = 53, mean age 52.3 ± 0.95 years) *India* Note: Not possible to determine if HDL= low in both groups due to lack of gender stratification.	Not stated	Newly diagnosed T2DM: TC (mg/dL): 160.37 ± 5.05 (mmol/L): 4.15 ± 0.13 TG (mg/dL): 156.08 ± 10.83 * (*high*) (mmol/L): 1.76 ± 0.12 LDL-C (mg/dL): 87.38 ± 5.18 (mmol/L): 2.26 ± 0.13 HDL-C (mg/dL): 43.73 ± 1.29 (*low?)* (mmol/L): 1.13 ± 0.03 AIP: 0.16 ± 0.03 * (*medium*) Controls: TC (mg/dL): 153.67 ± 4.70 (mmol/L): 3.97 ± 0.12 TG (mg/dL): 126.13 ± 4.95 * (mmol/L): 1.42 ± 0.06 LDL-C (mg/dL): 84.08 ± 4.48 (mmol/L): 2.17 ± 0.12 HDL-C (mg/dL): 44.37 ± 0.49 (*low?)* (mmol/L): 1.15 ± 0.01 AIP: 0.08 ± 0.01* (*low*)
**Agrawall et al., 2014**	400 adults newly diagnosed with T2DM; Urban controls (*n* = 100, mean age 50.3 ± 10.4, 50 males/50 females) Urban diabetics (*n* = 100, mean age 52.9 ± 9.2, 61 males/39 females) Rural controls (*n* = 100, mean age 49.3 ± 11.0 years, 51 males/49 females) Rural diabetics (*n* = 100, mean age 51.5 ± 10.0 years, 47 males, 53 females) Note: *p*-values are between the two urban and between the two rural groups. Note: Not possible to determine if HDL= low groups 1, 3, and 4 due to lack of gender stratification. *India*	Retrospective report	Urban controls: TC (mg/dL): 202.54 ± 43.30 (mmol/L): 5.24 ± 1.12 TG (mg/dL): 157.63 ± 66.50 (*high*) (mmol/L): 1.78 ± 0.75 LDL-C (mg/dL): 126.09 ± 44.24 (mmol/L): 3.26 ± 1.14 HDL-C (mg/dL): 43.41 ± 6.34 (*low*?) (mmol/L): 1.12 ± 0.16 AIP: 0.53 ± 0.18 (*high*) Urban diabetics: TC (mg/dL): 219.05 ± 51.56 * (mmol/L): 5.66 ± 1.33 TG (mg/dL): 218.65 ± 85.38 * (*high*) (mmol/L): 2.47 ± 0.96 LDL-C (mg/dL): 132.64 ± 52.99 (mmol/L): 3.43 ± 1.37 HDL-C (mg/dL): 38.15 ± 8.35 * (*low*) (mmol/L): 0.99 ± 0.22 AIP: 0.73 ± 0.20* (*high*) Rural controls: TC (mg/dL): 199.74 ± 44.19 (mmol/L): 5.17 ± 1.14 TG (mg/dL): 157.33 ± 67.62 (*high*) (mmol/L): 1.78 ± 0.76 LDL-C (mg/dL): 126.31 ± 44.59 (mmol/L): 3.27 ± 1.15 HDL-C (mg/dL): 44.06 ± 6.65 (*low*?) (mmol/L): 1.14 ± 0.17 AIP: 0.52 ± 0.17 (*high*) Rural diabetics: TC (mg/dL): 222.46 ± 56.62 * (mmol/L): 5.75 ± 1.46 TG (mg/dL): 215.42 ± 84.48 * (*high*) (mmol/L): 2.43 ± 0.95 LDL-C (mg/dL): 131.21 ± 50.12 (mmol/L): 3.39 ± 1.30 HDL-C (mg/dL): 45.30 ± 6.70 (*low*?) (mmol/L): 1.17 ± 0.17 AIP: 0.65 ± 0.19 * (*high*)
**Guzel et al., 2021**	451 patients with chronic total occlusion (100% stenosis); Poor collateral (*n* = 232, mean age 60.70 ± 10.85, 70.3% male) Good collateral (*n* = 219, mean age 68.86 ± 11.40, 74% male) Note: Collateral based on angiography results. Note: Not possible to determine if HDL-C = low in both groups due to lack of gender stratification.	Not stated	Poor collateral: TC (mg/dL): 183.9 ± 47.95 * (mmol/L): 4.76 ± 1.24 TG (mg/dL): 186.7 ± 86.75 ** (*high*) (mmol/L): 2.11 ± 0.98 LDL-C (mg/dL): 102.1 ± 39.12 (mmol/L): 2.64 ± 1.01 HDL-C (mg/dL): 41.77 ± 26.73 (low?) (mmo/L): 1.08 ± 0.69 AIP: 0.63 ± 0.25 ** (*high*) Good collateral: TC (mg/dL): 173.4 ± 46.63 * (mmol/L): 4.48 ± 1.21 TG (mg/dL): 143.9 ± 69.16 ** (mmol/L): 1.62 ± 0.78 LDL-C (mg/dL): 102.4 ± 38.41 (mmol/L): 2.65 ± 0.99 HDL-C (mg/dL): 43.38 ± 8.71 (mmol/L): 1.12 ± 0.23 AIP: 0.48 ± 0.25 ** (*high*)
**Krivosikova et al., 2015**	411 apparently healthy adults; 0 CMR factors (*n* = 162, mean age 30.0 ± 8.9 years, 20% males) 1–2 CMR factors (*n* = 162, mean age 36.2 ± 13.9 years, 46% males) 3–4 CMR factors (*n* = 87, mean age 46.1 ± 14.6, 72% males) *Slovakia* Note: *p*-values compared to 0 CMR factors group.	Cross-sectional study	0 CMR factors: TC (mmol/L): 4.4 ± 0.8 (mg/dL): 170.15 ± 30.94 TG (mmol/L): 0.8 ± 0.3 (mg/dL): 70.86 ± 26.57 LDL-C (mmol/L): 2.4 ± 0.7 (mg/dL): 92.81 ± 27.07 HDL-C (mmol/L): 1.5 ± 0.4 (mg/dL): 58.0 ± 15.47 AIP: −0.30 ± 0.21 (*low*) 1–2 CMR factors: TC (mmol/L): 4.6 ± 0.9 ** (mg/dL): 177.88 ± 34.8 TG (mmol/L): 1.2 ± 0.7 *** (mg/dL): 106.29 ± 62.00 LDL-C (mmol/L): 2.7 ± 0.8 ** (mg/dL): 104.41 ± 30.94 HDL-C (mmol/L): 1.4 ± 0.4 *** (mg/dL): 54.14 ± 15.47 AIP: −0.10 ± 0.30 *** (*low*) 3–4 CMR factors: TC (mmol/L): 5.3 ± 0.8 *** (mg/dL): 204.95 TG (mmol/L): 2.2 ± 1.1 *** (*high*) (mg/dL): 194.86 ± 97.43 LDL-C (mmol/L): 3.2 ± 0.7 *** (mg/dL): 123.74 ± 27.07 HDL-C (mmol/L): 1.1 ± 0.2 *** (*low*?) (mg/dL): 42.54 ± 7.73 AIP: 0.28 ± 0.24 (*high*)
**Li et al., 2015**	1772 clinically suspected CAD cases; CAD diagnosed (≥50% coronary lesion, *n* = 1057, mean age 58.03 ± 10.0 years, 68.4% males) CAD not diagnosed (*n* = 715, mean age 52.56 ± 11.67 years, 49.5% males) *China*	Cross-sectional study	CAD diagnosed: TC (mmol/L): 4.92 ± 0.99 (mg/dL): 190.26 ± 30.28 TG (mmol/L): 1.78 *** (range: 1.29–2.43) (*high*) (md/dL): 157.66 (range: 49.88–93.97) LDL-C (mmol/L): 3.22 ± 0.91 * (mg/dL): 124.52 ± 35.19 HDL-C (mmol/L): 1.09 ± 0.29 *** (*low*?) (mg/dL): 42.15 ± 11.21 AIP: 0.23 ± 0.26 *** (*high*) CAD not diagnosed: TC (mmol/L): 4.87 ± 1.03 (mg/dL) 188.32 ± 39.83 TG (mmol/L): 1.53 *** (range: 1.05–2.18) (mg/dL): 135.52 (range: 40.6–84.3) LDL-C (mmol/L): 3.12 ± 0.96 * (mg/dL): 120.65 ± 37.12 HDL-C (mmol/L): 1.19 ± 0.36 *** (*low*?) (mg/dL): 46.02 ± 13.92 AIP: 0.13 ± 0.33 *** (*medium*)
**Cibickova et al., 2019**	685 asymptomatic dyslipidemic subjects; Hypertriglyceridemic waist present (*n* = 202, mean age 49.4 ± 11.8 *** years) Hypertriglyceridemic waist absent (*n* = 483, mean age 43.5 ± 15.1 *** years) Risky waist present (*n* = 185, mean age 51.5 ± 11.5 *** years) Risky waist absent (*n* = 501, mean age 43.0 ± 14.7 *** years) BMI > 25 kg/m^2^ (*n* = 382, mean age 48.6 ± 12.4 *** years) BMI < 25 kg/m^2^ (*n* = 303, mean age 41.0 ± 15.7 *** years) *Czech Republic*	Cross-sectional study	Hypertriglyceridemic waist present: TC (mmol/L): 7.32 ± 2.44 (mg/dL): 283.06 ± 94.35 TG (mmol/L): 5.31 ± 5.27 (*high*) (mg/dL): 470.33 ± 466.78 LDL-C (mmol/L): 4.07 ± 2.06 (mg/dL): 157.39 ± 79.66 HDL-C (mmol/L): 1.18 ± 0.41 (*low*?) (mg/dL): 45.63 ± 15.85 AIP: 0.57 ± 0.34 (*high*) Hypertriglyceridemic waist absent: TC (mmol/L): 6.10 ± 1.62 (mg/dL): 235.89 ± 62.65 TG (mmol/L): 1.65 ± 1.93 (mg/dL): 146.15 ± 170.95 LDL-C (mmol/L): 3.78 ± 1.35 (mg/dL): 146.17 ± 52.2 HDL-C (mmol/L): 1.59 ± 0.44 (mg/dL): 61.48 ± 17.01 AIP: −0.05 ± 0.30 (*low*) Risky waist present: TC (mmol/L): 6.97 ± 2.42 (mg/dL): 268.52 ± 93.58 TG (mmol/L): 4.04 ± 5.21 (*high*) (mg/dL): 357.84 ± 461.47 LDL-C (mmol/L): 4.05 ± 1.89 (mg/dL): 156.61 ± 73.09 HDL-C (mmol/L): 1.32 ± 0.45 (mg/dL): 51.04 ± 17.4 AIP: 0.35 ± 0.41 (*high*) Risky waist absent: TC (mmol/L): 6.27 ± 1.76 (mg/dL): 242.46 ± 68.06 TG (mmol/L): 2.24 ± 2.79 (*high*) (mg/dL): 198.41 ± 247.12 LDL-C (mmol/L): 3.79 ± 1.46 (mg/dL): 146.56 ± 56.46 HDL-C (mmol/L): 1.52 ± 0.47 (mg/dL): 58.78 ± 18.17 AIP: 0.05 ± 0.40 (*low*) BMI > 25 kg/m^2^ TC (mmol/L): 6.73 ± 2.26 (mg/dL): 260.25 ± 87.39 TG (mmol/L): 3.57 ± 4.57 (*high*) (mg/dL): 316.21 ± 404.78 LDL-C (mmol/L): 3.97 ± 1.72 (mg/dL): 153.52 ± 66.51 HDL-C (mmol/L): 1.31 ± 0.41 (mg/dL): 50.66 ± 15.85 AIP: 0.30 ± 0.42 (*high*) BMI < 25 kg/m^2^ TC (mmol/L): 6.13 ± 1.50 (mg/dL): 237.05 ± 58.00 TG (mmol/L): 1.67 ± 1.90 (mg/dL): 147.92 ± 168.29 LDL-C (mmol/L): 3.73 ± 1.40 (mg/dL): 144.24 ± 54.14 HDL-C (mmol/L): 1.67 ± 0.47 (mg/dL): 64.58 ± 18.17 AIP: −0.08 ± 0.33 (*low*)
**Vaverkova et al., 2017**	607 asymptomatic dyslipidemic patients (mean age 45.6 ± 14.0 years, 295 males/312 females) *Czech Republic*	Not stated	TC (mmol/L): 6.73 ± 1.64 (mg/dL): 260.25 ± 63.42 TG (mmol/L): 1.91 (range: 1.32–2.98) (*high*) (mg/dL): 169.18 (range: 51.04–115.24) LDL-C (mmol/L): 4.17 ± 1.28 (mg/dL): 161.25 ± 49.5 HDL-C (mmol/L): 1.44 ± 0.45 (mg/dL): 55.68 ± 17.4 AIP: 0.13 (*medium*)
**Al-Bazi et al., 2011**	106 healthy adults age range 21–45 years; Female non-smokers (mean age 36.3 ± 12.1 years) Male non-smokers (mean age 34.3 ± 12.7 years) Female smokers (mean age 34.6 ± 7.5 years) Male smokers (mean age 35.1 ± 12.7) *Saudi Arabia* Note: Total non-smokers *n* = 51; total smokers *n* = 55. *p*-values between females and males.	Not stated	Female non-smokers: TC (mmol/L): 5.10 ± 0.90 * (mg/dL): 197.22 ± 34.8 TG (mmol/L): 1.69 ± 0.54 ** (mg/dL): 149.69 ± 47.83 LDL-C (mmol/L): 2.95 ± 0.33 *** (mg/dL): 114.08 ± 12.76 HDL-C (mmol/L): 1.39 ± 0.31 *** (mg/dL): 53.75 ± 11.99 AIP: 0.09 ± 0.05 *** (*low*) Male non-smokers: TC (mmol/L): 4.90 ± 0.80 (mg/dL): 189.48 ± 30.94 TG (mmol/L): 1.78 ± 0.42 *** (*high*) (mg/dL): 157.66 ± 37.20 LDL-C (mmol/L): 3.03 ± 0.41 (mg/dL): 117.17 ± 15.85 HDL-C (mmol/L): 1.29 ± 0.29 * (mg/dL): 49.88 ± 11.21 AIP: 0.13 ± 0.09*** (*medium*) Female smokers: TC (mmol/L): 4.10 ± 0.90 * (mg/dL): 158.55 ± 34.8 TG (mmol/L): 2.02 ± 0.44 ** (*high*) (mg/dL): 178.92 ± 38.97 LDL-C (mmol/L): 2.02 ± 0.46 *** (mg/dL): 78.11 ± 17.79 HDL-C (mmol/L): 0.90 ± 0.12 *** (*low*) (mg/dL): 34.8 ± 4.64 AIP: 0.35 ± 0.13 *** (*high*) Male non-smokers: TC (mmol/L): 4.70 ± 1.13 (mg/dL): 181.75 ± 43.7 TG (mmol/L): 2.26 ± 0.51 *** (*high*) (mg/dL): 200.18 ± 45.17 LDL-C (mmol/L): 3.11 ± 0.61 (mg/dL): 120.26 ± 23.59 HDL-C (mmol/L): 1.03 ± 0.26 * (mg/dL): 39.83 ± 10.05 AIP: 0.34 ± 0.11 *** (*high*)
**Wang et al., 2020**	3600 suspected CAD cases divided based on SYNTAX score; Non-CAD (*n* = 1347, mean age 58.7 ± 8.9 years, 62.3% males) 1 ≤ SS ≤ 23 (*n* = 1448, mean age 61.7 ± 10.7 years, 61.3% males) 23 ≤ SS ≤ 33 (*n* = 569, mean age 61.2 ± 10.4 years, 61.9% males) SS ≥ 33 (*n* = 236, mean age 60.5 ± 11.4 years, 69.9% males) *China* Note: *p*-value compared to non-CAD group.	Not stated	Non-CAD TC (mmol/L): 3.9 ± 0.8 (mg/dL): 150.81 ± 30.94 TG (mmol/L): 1.3 ± 0.6 (mg/dL): 115.15 ± 53.14 LDL-C (mmol/L): 2.3 ± 0.7 (mg/dL): 88.94 ± 27.07 HDL-C (mmol/L): 1.3 ± 0.3 (mg/dL): 50.27 ± 27.07 AIP: 1.9 ± 0.2 (*high*) 1 ≤ SS ≤ 23 TC (mmol/L): 4.2 ± 1.1 ** (mg/dL): 162.41 ± 42.54 TG (mmol/L): 1.7 ± 0.9 ** (*high*) (mg/dL): 150.58 ± 79.72 LDL-C (mmol/L): 2.6 ± 0.9 ** (mg/dL): 100.54 ± 34.8 HDL-C (mmol/L): 1.2 ± 0.2 ** (*low*?) (mg/dL): 46.4 ± 7.73 AIP: 2.1 ± 0.2 ** (*high*) 23 ≤ SS ≤33 TC (mmol/L): 4.4 ± 1.2 ** (mg/dL): 170.15 ± 46.4 TG (mmol/L): 2.3 ± 1.4 ** (*high*) (mg/dL): 203.72 ± 124.00 LDL-C (mmol/L): 2.7 ± 0.8 ** (mg/dL): 104.41 ± 30.94 HDL-C (mmol/L): 1.1 ± 0.3 ** (*low*?) (mg/dL): 42.94 ± 11.6 AIP: 2.2 ± 0.2 (*high*) SS ≥ 33 TC (mmol/L): 4.9 ± 1.3 ** (mg/dL): 189.48 ± 50.27 TG (mmol/L): 2.5 ± 1.3 ** (mg/dL): 221.43 ± 115.15 LDL-C (mmol/L): 3.2 ± 1.1 ** (mg/dL): 123.74 ± 42.54 HDL-C (mmol/L): 1.0 ± 0.3 ** (mg/dL): 38.67 ± 11.6 AIP: 2.3 ± 0.3 **
**Yin et al., 2021**	4744 Chinese individuals with hypertension stratified based on AIP; Q1 (*n* = 1184, mean age 66.89 ± 9.11 years, 60.47% males) Q2 (*n* = 1187, mean age 65.67 ± 9.16 years, 48.53% males) Q3 (*n* = 1187, mean age 63.65 ± 9.17 years, 42.63% males) Q4 (*n* = 1186, mean age 61.68 ± 9.32 years, 47.89% males) *China*	Cross-sectional study	Q1: TC (mmol/L): 5.01 ± 1.01 *** (mg/dL): 193.74 ± 39.06 TG (mmol/L): 0.82 ± 0.21 *** (mg/dL): 72.63 ± 18.6 LDL-C (mmol/L): 2.57 ± 0.66 *** (mg/dL): 99.38 ± 25.52 HDL-C (mmol/L): 1.86 ± 0.40 *** (mg/dL): 71.93 ± 15.47 AIP: −0.35 ± 0.11 *** (*low*) Q2: TC (mmol/L): 5.10 ± 1.11 *** (mg/dL): 197.22 ± 42.92 TG (mmol/L): 1.22 ± 0.26 *** (mg/dL): 108.06 ± 23.03 LDL-C (mmol/L): 2.89 ± 0.77 *** (mg/dL): 111.76 ± 29.78 HDL-C (mmol/L): 1.55 ± 0.30 *** (mg/dL): 59.94 ± 11.6 AIP: −0.11 ± 0.05 *** (*low*) Q3: TC (mmol/L): 5.29 ± 1.08 *** (mg/dL): 204.56 ± 41.76 TG (mmol/L): 1.68 ± 0.36 *** (mmol/L): 148.80 ± 31.89 LDL-C (mmol/L): 3.15 ± 0.77 *** (mg/dL): 121.81 ± 29.78 HDL-C (mmol/L): 1.40 ± 0.27 *** (mg/dL): 54.14 ± 10.44 AIP: 0.08 ± 0.06*** (*low*) Q4: TC (mmol/L): 5.17 ± 1.17 *** (mg/dL): 199.92 ± 45.24 TG (mmol/L): 2.86 ± 0.92 *** (*high*) (mg/dL): 253.32 ± 81.49 LDL-C (mmol/L): 3.15 ± 0.80 *** (mg/dL): 121.81 ± 30.94 HDL-C (mmol/L): 1.20 ± 0.26 *** (*low*?) (mg/dL): 46.4 ± 10.05 AIP: 0.37 ± 0.14 *** (*high*)
**Choudhary et al., 2019**	615 normotensive (40.5%) AND never-treated subjects with primary hypertension 59.5%; T1 (*n* = 202, mean age 44.7 ± 12.3 years, 104 males/98 females) T2 (*n* = 208, mean age 44.1 ± 11.9 years, 106 males/102 females) T3 (*n* = 205, mean age 44.9 ± 11.6, 104 males/101 females) *Finland* Note: *p*-values compared to T1.	Cross-sectional study	T1: TC (mmol/L): 4.83 ± 1.0 (mg/dL): 186.77 ± 38.67 TG (mmol/L): 0.70 ± 0.21 (mg/dL): 62.00 ± 18.6 LDL-C (mmol/L): 2.70 ± 0.90 (mg/dL): 104.41 ± 34.80 HDL-C (mmol/L): 1.90 ± 0.40 (mg/dL): 73.74 ± 15.47 AIP: −0.44 ± 0.17 (*low*) T2: TC (mmol/L): 5.17 ± 1.0 * (mg/dL): 199.92 ± 38.67 TG (mmol/L): 1.09 ± 0.33 * (mg/dL): 168.29 ± 29.23 LDL-C (mmol/L): 3.12 ± 0.90 * (mg/dL): 120.65 ± 34.80 HDL-C (mmol/L): 1.58 ± 0.37 * (mg/dL): 61.1 ± 14.31 AIP: −0.17 ± 0.16 * (*low*) T3: TC (mmol/L): 5.44 ± 1.0 * (mg/dL): 210.36 ± 38.67 TG (mmol/L): 1.92 ± 0.92 * (*high*) (mg/dL): 170.06 ± 81.49 LDL-C (mmol/L): 3.40 ± 1.0 * (mg/dL): 131.48 ± 38.67 HDL-C (mmol/L): 1.30 ± 0.34 * (mg/dL): 50.27 ± 13.15 AIP: 0.15 ± 0.23 * (*medium*)
**Javardi et al., 2020**	157 individuals, age range 18–65 years, divided by weight; Healthy weight (*n* = 71, mean age 38.90 ± 10.976 years, 80.3% males) Overweight and obese (*n* = 86, mean age 38.60 ± 9.394 years, 81.6% males) *Iran*		Healthy weight: TC (mg/dL): 172.17 ± 35.30 (mmol/L): 4.45 ± 0.91 TG (mg/dL): 152.52 ± 85.38 (mmol/L): 1.72 ± 0.96 LDL-C (mg/dL): 98.44 ± 29.71 (mmol/L): 2.55 ± 0.77 HDL-C (mg/dL): 44.51 ± 8.41 (mmol/L): 1.15 ± 0.22 AIP: 0.170 ± 0.07 Overweight and obese: TC (mg/dL): 175.06 ± 29.90 (mmol/L): 4.53 ± 0.77 TG (mg/dL): 164.99 ± 84.67 (*high*) (mmol/L): 1.86 ± 0.96 LDL-C (mg/dL): 100.22 ± 25.75 (mmol/L): 2.59 ± 0.67 HDL-C (mg/dL): 42.40 ± 9.67 (*low*?) (mmol/L): 1.10 ± 0.25 AIP: 0.214 ± 0.111* (*high*)
**Mondal et al., 2021**	140 patients newly diagnosed with T2DM; 57.6% males; Over 70% primary/below primary education; stratified by gender for blood markers only *India*	Cross-sectional study	Males: TC (mg/dL): 193.58 ± 12.34 (mmol/L): 5.01 ± 0.32 TG (mg/dL): 148.41 ± 9.75 (mmol/L): 1.68 ± 0.11 LDL-C (mg/dL): 153.16 ± 14.97 (mmol/L): 3.96 ± 0.39 HDL-C (mg/dL): 40.52 ± 5.1 (mmol/L): 1.05 ± 0.13 AIP: 0.57 ± 0.07 (*high*) Females: TC (mg/dL): 193.44 ± 12.87 (mmol/L): 5.00 ± 0.33 TG (mg/dL): 149.97 ± 9.81 (mmol/L): 1.69 ± 0.11 LDL-C (mg/dL): 153.37 ± 15.43 (mmol/L): 3.97 ± 0.40 HDL-C (mg/dL): 40.07 ± 4.86 (*low*) (mmol/L): 1.04 ± 0.13 AIP: 0.57 ± 0.07 (*high*)
**Anandkumar et al., 2020**	60 apparently healthy young females, age range 18–30 years *India*	Cross-sectional study	TG (mg/dL): 130.42 ± 44.06 (mmol/L): 1.47 ± 0.50 HDL-C (mg/dL): 32.87 ± 6.03 (*low*) (mmol/L): 0.85 ± 0.16 AIP: 0.581 ± 0.148 (*high*)
**Nam et al., 2021**	3468 healthy Koreans, stratified based on AIP; Q1 (*n* = 868, mean age 50.3 ± 9.6 years, 242 males/626 females) Q2 (*n* = 878, mean age 52.3 ± 9.1 years, 417 males/461 females) Q3 (*n* = 880, mean age 52.9 ± 8.6 years, 606 males/274 females) Q4 (*n* = 842, mean age 51.6 ± 8.8 years, 705 males/137 females) *Korea*	Cross-sectional study	Q1: TC (mmol/L): 4.9 ± 0.8 (mg/dL): 189.48 ± 30.94 TG (mmol/L): 0.6 (range: 0.50–0.70) (mg/dL) 53.14 (range: 19.33–27.07) LDL-C (mmol/L): 2.9 ± 0.7 (mg/dL): 112.14 ± 27.07 HDL-C (mmol/L): 1.7 ± 0.3 (mg/dL): 65.74 ± 11.6 AIP: −0.04 ± 0.11 (*low*) Q2: TC (mmol/L): 4.9 ± 0.9 *** (mg/dL): 189.48 ± 34.8 TG (mmol/L): 0.9 *** (range: 0.80–1.00) (mg/dL): 79.72 (range: 30.94–38.67) LDL-C (mmol/L): 3.1 ± 0.8 *** (mg/dL): 119.88 ± 30.94 HDL-C (mmol/L): 1.4 ± 0.2 *** (mg/dL): 54.14 ± 7.73 AIP: −0.02 ± 0.06 *** (*low*) Q3: TC (mmol/L): 5.0 ± 0.9 *** (mg/dL): 193.35 ± 34.80 TG (mmol/L): 1.2 *** (range: 1.1–1.4) (mg/dL): 106.29 (range: 42.54–54.14) LDL-C (mmol/L): 3.2 ± 0.8 *** (mg/dL): 123.74 ± 30.94 HDL-C (mmol/L): 1.2 ± 0.2 *** (*low*?) (mg/dL): 46.40 ± 7.73 AIP: 0.02 ± 0.06 *** (*low*) Q4: TC (mmol/L): 5.1 ± 0.09 *** (mg/dL): 197.22 ± 3.48 TG (mmol/L): 2.0 *** (range: 1.7–2.5) (*high*) (mg/dL): 177.15 (range: 65.74–96.67) LDL-C (mmol/L): 3.3 ± 0.8 *** (mg/dL): 292.29 ± 70.86 HDL-C (mmol/L): 1.0 ± 0.2 *** (*low*?) (mg/dL): 38.67 ± 7.73 AIP: 0.32 ± 0.13 *** (*high*)
**Hosseini et al., 2020**	183 women; age range 20–35 years; stratified based on weight and metabolic state; Metabolically healthy normal weight (*n* = 53, mean age 26.36 ± 5.01 years) Normal weight obese (*n* = 29, mean age 27.21 ± 4.61 years) Metabolically healthy obese (*n* = 57, mean age 28.44 ± 4.53 years) Metabolically unhealthy obese (*n* = 44, mean age 27.09 ± 4.30 years) *Iran* Note: Normal weight obese = BMI <25 kg/m^2^ and body fat >30%. * = *p*-value between 1st and 3rd groups; ^ = *p*-value between 3rd and 4th groups.	Cross-sectional study	Metabolically healthy normal weight: TC (mg/dL): 174.13 ± 24.79 (mmol/L): 4.5 ± 0.64 TG (mg/dL): 100.52 ± 49.93 (mmol/L): 1.13 ± 0.56 LDL-C (mg/dL): 101.15 ± 23.66 (mmol/L): 2.62 ± 0.61 HDL-C (mg/dL): 52.18 ± 7.56 (mmol/L): 1.35 ± 0.20 AIP: 0.61 ± 0.19 (high) Normal weight obese: TC (mg/dL): 179.89 ± 29.15 (mmol/L): 4.65 ± 0.75 TG (mg/dL): 102.79 ± 64.52 (mmol/L): 1.16 ± 0.73 LDL-C (mg/dL): 107.20 ± 28.71 (mmol/L): 2.77 ± 0.74 HDL-C (mg/dL): 49.10 ± 7.59 (*low*?) (mmol/L) 1.27 ± 0.20 AIP: 0.63 ± 0.24 (*high*) Metabolically healthy obese: TC (mg/dL): 180.86 ± 30.31 (mmol/L): 4.68 ± 0.78 TG (mg/dL): 100.94 ± 33.06 ^^^ (mmol/L): 1.14 ± 0.37 LDL-C (mg/dL): 113.80 ± 28.28 (mmol/L): 2.94 ± 0.73 HDL-C (mg/dL): 47.77 ± 9.13 ***^^^ (*low*?) (mmol/L): 1.24 ± 0.24 AIP: 0.67 ± 0.17 ^^^ (*high*) Metabolically unhealthy obese: TC (mg/dL): 187.45 ± 36.18 (mmo/L): 4.85 ± 0.94 TG (mg/dL): 157.50 ± 47.96 ^^^ (*high*) (mmol/L): 1.78 ± 0.54 LDL-C (mg/dL): 120.81 ± 34.83 (mmol/L): 3.12 ± 0.90 HDL-C (mg/dL): 37.79 ± 6.04 ^^^ (*low*) (mmol/L): 0.98 ± 0.16 AIP: 0.96 ± 0.17 ^^^ (*high*)
**Cakirca and Celik, 2019**	225 diabetic subjects stratified based on HbA1c; HbA1c < 7% (*n* = 59, mean age 56 years, 20.3% men) HbA1c 7–9% (*n* = 71, mean age 59 years, 29.6% men) HbA1c >9% (*n* = 95, mean age 56 years, 32.6% men) *Turkey* Note: *p*-values compared to the lowest HbA1c group.	Retrospective study	HbA1c < 7% TC (mg/dL): 218.5 ± 43.8 (mmol/L): 5.65 ± 1.13 TG (mg/dL): 132.1 (range: 69.30–358.20) (mmol/L): 1.48 (range: 1.79—9.26) LDL-C (mg/dL): 139.9 ± 39.5 (mmol/L): 3.62 ± 1.02 HDL-C (mg/dL): 44.9 (range: 26.80–72.9) (*low*?) (mmol/L): 1.16 (range: 0.69–1.89) AIP: 0.50 ± 0.27 (*high*) HbA1c 7–9% TC (mg/dL): 204.8 ± 52.1 (mmol/L): 5.3 ± 1.35 TG (mg/dL): 142.5 (range: 51.5–490.6) (mmol/L): 1.61 (range: 1.33–12.69) LDL-C (mg/dL): 128.7 ± 43.1 (mmol/L): 3.33 ± 1.11 HDL-C (mg/dL): 38.6 * (range: 24.00–77.60) (*low*) (mmol/L): 1.00 (range: 0.62–2.01) AIP: 0.58 ± 0.30 (*high*) HbA1c > 9% TC (mg/dL): 214.4 ± 46.6 (mmol/L): 5.54 ± 1.21 TG (mg/dL): 189.8 (range: 59.2–603.80) (*high*) (mmol/L): 2.14 (range: 1.53–15.61) LDL-C (mg/dL): 130.8 ± 40.6 (mmol/L): 3.38 ± 1.06 HDL-C (mg/dL): 38.8 *** (range: 22.40–72.00) (*low*) (mmol/L): 1.00 (range: 0.58–1.86) AIP: 0.68 ± 0.29 *** (*high*)
**Lwow and Bohdanowicz-Pawlak, 2020**	318 post-menopausal Polish women of Caucasian origin (mean age 55.3 ± 2.8 years) *Poland*	Not stated	TC (mg/dL): 245.00 ± 42.00 (mmol/L): 6.34 ± 1.09 TG (mg/dL): 106.70 ± 46.80 (mmol/L): 1.20 ± 0.53 LDL-C (mg/dL): 153.00 ± 38.00 (mmol/L): 3.96 ± 0.98 HDL-C (mg/dL): 70.5 ± 17.4 (mmol/L): 1.82 ± 0.45 AIP: 0.35 ± 0.58 (*high*)
**Wang et al., 2016**	1475 adults stratified by gender; Men (*n* = 829, mean age 40 years ***) Women (*n* = 646, mean age 38 years ***)	Not stated	Men: TC (mmol/L): 4.78 ± 0.85 (mg/dL): 184.84 ± 32.87 TG (mmol/L): 1.39 (range: 0.59–4.35) (mg/dL): 123.12 (range: 22.82–168.21) LDL-C (mmol/L): 3.12 ± 0.80 (mg/dL): 120.65 ± 30.94 HDL-C (mmol/L): 1.18 (range: 0.85–1.74) (mg/dL): 45.63 (range: 32.87–67.29) AIP: 0.10 ± 0.29 (*medium*) Women: TC (mmol/L): 4.76 ± 0.86 (mg/dL): 184.07 ± 33.26 TG (mmol/L): 0.93 (range: 0.47–2.49) (mg/dL): 82.37 (range: 18.17–96.29) LDL-C (mmol/L): 2.86 ± 0.76 (mg/dL): 110.60 ± 29.39 HDL-C (mmol/L): 1.47 (range: 1.01–2.18) (mg/dL): 56.84 (range: 39.06–84.30) AIP: −0.18 ± 0.28 (*low*)
**Al-Shaer et al., 2021**	140 T2DM patients; T2DM with CAD (*n* = 70) T2DM without CAD (*n* = 70) *Egypt* Note: No units stated for LDL.		T2DM with CAD: TG (mmol/L): 12.46 ± 4.10 ** (*high*) (mg/dL): 1103.63 ± 363.15 LDL-C: 156.17 ± 17.9 ** HDL-C (mmol/L): 1.82 ± 0.59 (mg/dL): 70.38 ± 22.82 AIP: 1.02 ± 0.30 ** (*high*) T2DM without CAD: TG (mmol/L): 5.21 ± 1.8 ** (*high*) (mg/dl): 461.47 ± 159.43 LDL-C: 126.82 ± 12.91 ** HDL-C (mmol/L): 1.98 ± 0.67 (mg/dL): 76.57 ± 25.91 AIP: 0.01 ± 0.21 ** (*low*)
**Nwagha et al., 2010**	80 females; Apparently healthy post-menopausal women (*n*= 50, age range 50–70 years) Apparently healthy pre-menopausal women (*n* = 30, age range 25–49 years) *Nigeria*		Apparently healthy post-menopausal women: TG (mmol/L): 1.68 ± 0.63 (mg/dL): 148.8 ± 55.80 LDL-C (mmol/L): 4.46 ± 0.68 (mg/dL): 172.47 ± 26.3 HDL-C (mmol/L): 1.19 ± 0.28 (*low*) (mg/dL): 46.02 ± 10.83 AIP: 0.25 ± 0.35 (*high*) Apparently healthy pre-menopausal women: TG (mmol/L): 1.02 ± 0.44 (mg/dL): 90.35 ± 38.97 LDL-C (mmol/L): 2.71 ± 1.13 (mg/dL): 104.08 ± 43.70 HDL-C (mmol/L): 1.52 ± 0.36 (mg/dL): 58.78 ± 13.92 AIP: −0.17 ± 0.09 (*low*)

* *p* ≤ 0.05, ** *p* ≤ 0.01, *** *p* ≤ 0.001, ^ = the *p*-value between groups.

**Table 5 healthcare-11-00966-t005:** Outline of the studies interlinking blood glucose data and AIP. Author, year, subject characteristics, study design, primary outcomes, and other study outcomes of interest (HbA1c, T2DM, blood glucose, insulin) are presented for each study where available [33,37,39,43,44,50,52].

Author and Year	Population Characteristics and Ethnicity/Provenance	Design	Study Outcomes
**Hanamatsu et al., 2014**	23 students of 18–27 years; Obese (*n* = 12, mean age 21.7 ± 2.8 years, 12 females/0 males) Healthy weight (*n* = 11, mean age 23.2 ± 2.4, 7 males/4 females) *Japan*	Not stated	Obese: HbA1c (%): 5.1 ± 0.3 AIP: 0.81 ± 0.28 ** (*high*) Healthy weight control: HbA1c (%): 5.1 ± 0.4 AIP: 0.30 ± 0.26 ** (*high*)
**Guzel et al., 2021**	451 patients with chronic total occlusion (100% stenosis); Poor collateral (*n* = 232, mean age 60.70 ± 10.85, 70.3% male) Good collateral (*n* = 219, mean age 68.86 ± 11.40, 74% male) Note: Collateral based on angiography results. *Turkey*	Not stated	Poor collateral: T2DM: 45.3% ** AIP: 0.63 ± 0.25 ** (*high*) Good collateral: T2DM: 25.1% ** AIP: 0.48 ± 0.25 ** (*high*)
**Li et al., 2015**	1772 clinically suspected CAD cases; CAD diagnosed (≥50% coronary lesion, *n* = 1057, mean age 58.03 ± 10.0 years, 68.4% males) CAD not diagnosed (*n* = 715, mean age 52.56 ± 11.67 years, 49.5% males) *China*	Cross-sectional study	CAD diagnosed: T2DM: 26.6% *** AIP: 0.23 ± 0.26 (*high*) CAD not diagnosed: T2DM: 10.1% *** AIP: 0.13 ± 0.33 (*medium*)
**Wang et al., 2020**	3600 suspected CAD cases divided based on SYNTAX score; Non-CAD (*n* = 1347, mean age 58.7 ± 8.9 years, 62.3% males) 1 ≤ SS ≤ 23 (*n* = 1448, mean age 61.7 ± 10.7 years, 61.3% males) 23 ≤ SS ≤ 33 (*n* = 569, mean age 61.2 ± 10.4 years, 61.9% males) SS ≥ 33 (*n* = 236, mean age 60.5 ± 11.4 years, 69.9% males) *China* Note: *p*-value compared to non-CAD group.	Not stated	Non-CAD T2DM: 17.5% AIP: 1.9 ± 0.2 (*high*) 1 ≤ SS ≤ 23 T2DM: 26.3% *** AIP: 2.1 ± 0.2 *** (*high*) 23 ≤ SS ≤ 33 T2DM: 28.6% *** AIP: 2.2 ± 0.2 *** (*high*) SS ≥ 33 T2DM: 39% *** AIP: 2.3 ± 0.3 *** (*high*)
**Yin et al., 2021**	4744 Chinese individuals with hypertension stratified based on AIP; Q1 (*n* = 1184, mean age 66.89 ± 9.11 years, 60.47% males) Q2 (*n* = 1187, mean age 65.67 ± 9.16 years, 48.53% males) Q3 (*n* = 1187, mean age 63.65 ± 9.17 years, 42.63% males) Q4 (*n* = 1186, mean age 61.68 ± 9.32 years, 47.89% males) Note: T2DM is self-reported. *China*	Cross-sectional study	Q1: T2DM: 4.39% *** AIP: −0.35 ± 0.11 *** (*low*) Q2: T2DM: 6.23% *** AIP: −0.11 ± 0.05 *** (*low*) Q3: T2DM: 9.10% *** AIP: 0.08 ± 0.06 *** (*low*) Q4: T2DM: 11.72% *** AIP: 0.37 ± 0.14 *** (*high*)
**Hosseini et al., 2020**	183 women; age range 20–35 years; stratified based on weight and metabolic state; Metabolically healthy normal weight (*n* = 53, mean age 26.36 ± 5.01 years) Normal weight obese (*n* = 29, mean age 27.21 ± 4.61 years) Metabolically healthy obese (*n* = 57, mean age 28.44 ± 4.53 years) Metabolically unhealthy obese (*n* = 44, mean age 27.09 ± 4.30 years) *Iran* Note: Normal weight obese = BMI <25 kg/m^2^ and body fat >30%. * = *p*-value between 1st and 3rd groups; ^ = *p*-value between 3rd and 4th groups.	Cross-sectional study	Metabolically healthy normal weight: Insulin (μu): 9.70 ± 5.07 AIP: 0.61 ± 0.19 (*high*) Normal weight obese: Insulin (μu): 10.69 ± 5.96 AIP: 0.63 ± 0.24 (*high*) Metabolically healthy obese: Insulin (μu): 11.32 ± 6.52 AIP: 0.67 ± 0.17 *** (*high*) Metabolically unhealthy obese: Insulin (μu): 12.61 ± 7.96 AIP: 0.96 ± 0.17 *** (*high*)
**Lwow and Bohdanowicz-Pawlak, 2020**	318 post-menopausal Polish women of Caucasian origin (mean age 55.3 ± 2.8 years) *Poland*	Not stated	Glucose (mM): 4.97 ± 0.69 Insulin (μIU/mL): 6.8 ± 3.5 AIP: 0.35 ± 0.58 (high)

* *p* ≤ 0.05, ** *p* ≤ 0.01, *** *p* ≤ 0.001, ^ = the *p*-value between groups.

**Table 6 healthcare-11-00966-t006:** Outline of the studies interlinking blood pressure and AIP. Author, year, subject characteristics, study design, primary outcomes, and other study outcomes of interest (SBP, DBP, and % of participants suffering from hypertension) are presented for each study where available [38,39,43,45,50,54].

Author and Year	Population Characteristics and Ethnicity/Provenance	Design	Study Outcomes
**Krivosikova et al., 2015**	411 apparently healthy adults; 0 CMR factors (*n* = 162, mean age 30.0 ± 8.9 years, 20% males) 1–2 CMR factors (*n* = 162, mean age 36.2 ± 13.9 years, 46% males) 3–4 CMR factors (*n* = 87, mean age 46.1 ± 14.6, 72% males) *Slovakia*	Cross-sectional study	0 CMR factors: SBP (mmHg): 114.4 ± 7.5 DBP (mmHg): 72.0 ± 6.5 AIP: −0.30 ± 0.21 (*low*) 1–2 CMR factors: SBP (mmHg): 127.2 ± 13.3 DBP (mmHg): 78.0 ± 7.8 AIP: −0.10 ± 0.30 (*low*) 3–4 CMR factors: SBP (mmHg): 137.6 ± 13.1 (*high*) DBP (mmHg): 83.4 ± 7.6 AIP: 0.28 ± 0.24 (*high*)
**Li et al., 2015**	1772 clinically suspected CAD cases; CAD diagnosed (≥50% coronary lesion, *n* = 1057, mean age 58.03 ± 10.0 years, 68.4% males) CAD not diagnosed (*n* = 715, mean age 52.56 ± 11.67 years, 49.5% males) *China*	Cross-sectional study	CAD diagnosed: Hypertension: 63.5% *** AIP: 0.23 ± 0.26 *** (*high*) CAD not diagnosed: Hypertension: 43.2% *** AIP: 0.13 ± 0.33 *** (*medium*)
**Wang et al., 2020**	3600 suspected CAD cases divided based on SYNTAX score; Non-CAD (*n* = 1347, mean age 58.7 ± 8.9 years, 62.3% males) 1 ≤ SS ≤ 23 (*n* = 1448, mean age 61.7 ± 10.7 years, 61.3% males) 23 ≤ SS ≤ 33 (*n* = 569, mean age 61.2 ± 10.4 years, 61.9% males) SS ≥ 33 (*n* = 236, mean age 60.5 ± 11.4 years, 69.9% males) *China*	Not stated	Non-CAD: Hypertension: 17.5% *** AIP: 1.9 ± 0.2 *** 1 ≤ SS ≤ 23 Hypertension: 55.6%*** AIP: 2.1 ± 0.2 *** 23 ≤ SS ≤ 33 Hypertension: 57.8% *** AIP: 2.2 ± 0.2 *** SS ≥ 33 Hypertension: 56.8% *** AIP: 2.3 ± 0.3 ***
**Choudhary et al., 2019**	615 normotensive (40.5%) AND never-treated subjects with primary hypertension 59.5%); T1 (*n* = 202, mean age 44.7 ± 12.3 years, 104 males/98 females) T2 (*n* = 208, mean age 44.1 ± 11.9 years, 106 males/102 females) T3 (*n* = 205, mean age 44.9 ± 11.6, 104 males/101 females) *Finland* Note: *p*-values compared to T1.	Cross-sectional study	T1: SBP (mmHg): 135.7 ± 19.8 (*high*) DBP (mmHg): 86.1 ± 12.1 (*high*) AIP: −0.44 ± 0.17 (*low*) T2: SBP (mmHg): 140.2 ± 20.3 (*high*) DBP (mmHg): 89.5 ± 12.7 (*high*) AIP: −0.17 ± 0.16 (*low*) T3: SBP (mmHg): 145.7 ± 20.7 (*high*) DBP (mmHg): 93.0 ± 11.4 (*high*) AIP: 0.15 ± 0.23 (*low*)
**Hosseini et al., 2021**	183 women; age range 20–35 years; stratified based on weight and metabolic state; Metabolically healthy normal weight (*n* = 53, mean age 26.36 ± 5.01 years) Normal weight obese (*n* = 29, mean age 27.21 ± 4.61 years) Metabolically healthy obese (*n* = 57, mean age 28.44 ± 4.53 years) Metabolically unhealthy obese (*n* = 44, mean age 27.09 ± 4.30 years) *Iran* Note: Normal weight obese = BMI < 25 kg/m^2^ and body fat >30%. * = *p*-value between 1st and 3rd groups; ^ = *p*-value between 3rd and 4th groups.	Cross-sectional study	Metabolically healthy normal weight: SBP (mmHg): 113.77 ± 11.09 DBP (mmHg): 75.94 ± 9.96 AIP: 0.61 ± 0.19 *** (*high*) Normal weight obese: SBP (mmHg): 115.70 ± 11.27 DBP (mmHg): 76.93 ± 10.81 AIP: 0.63 ± 0.24 (*high*) Metabolically healthy obese: SBP (mmHg): 115.37 ± 14.31 ^^^ DBP (mmHg): 73.03 ± 15.08 ^ AIP: 0.67 ± 0.17 (*high*) Metabolically unhealthy obese: SBP (mmHg): 129.28 ± 14.18 ^^^ DBP (mmHg): 81.83 ± 13.74 ^ AIP: 0.96 ± 0.17 (*high*)
**Al-Shaer et al., 2021**	140 T2DM patients; T2DM with CAD (*n* = 70) T2DM without CAD (*n* = 70) *Egypt*	Not stated	T2DM with CAD: Hypertension: 64.3% AIP: 1.02 ± 0.31 (*high*) T2DM without CAD: Hypertension: 61.4% AIP: 0.01 ± 0.21 (*low*)

* *p* ≤ 0.05, ** *p* ≤ 0.01, *** *p* ≤ 0.001, ^ = the *p*-value between groups.

## Data Availability

Not applicable.

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
