# Peer review of "The Association between the Atherogenic Index of Plasma and Cardiometabolic Risk Factors: A Review"

_healthcare, 2023, doi:10.3390/healthcare11070966_

Round 1
Reviewer 1 Report
The aim of the work is clearly formulated and contains convincing justification. Materials and methods – raises no objections. Study selection and inclusion/exclusion criteria – described in detail and clearly with strengths and limitations. The results of our current review are presented in tables very comprehensive and legible. Interpretation–interesting and insightful. It's hard not to agree with the authors of limitations and necessity future research.
The table does not contain information whether patients in the studies included in the analysis take hypolipemic drugs, which may significantly affect the value of AIP.
The AIP index seems to be an important parameter in estimating the risk of cardiovascular events. Currently, nHDL is also such an indicator - there is no discussion about this parameter.
Lipid profile values are missing in some tables. In combination with AIP, it would be good to supplement.
Reviewer 2 Report
The article has an interesting topic, metabolic syndrome. I would recommend a figure for this review article a figure to illustrate the relationship between MetS, type 2 diabtes, and CVD.
Reviewer 3 Report
The authors provided a comprehensive literature review on the association between the atherogenic index of plasma and cardiometabolic risk factors. The manuscript is valuable and clearly written. The adressed topic is interesting and important as it may have clinical implications. I have several minor concerns.
• The article is very long and the authors presented much data. In my opinion it makes it difficult to read.
• The sentence in the abstract (lines 24-26) is not clear.
• Please explain all of the abbreviations - CAD.
• Please add units in all of the missing places in the tables( mean age-years, SBP, DBP- )
• Why there is a question mark next to the WhPR in Table 3?
• I suggest to add mg/dl along with mmol/l. Some cholesterol and triglycerides results are presented as mmol/l whereas the others as mg/dl. It is misleading
•line 423- thereis a spelling mistake - by instead of my?
Reviewer 4 Report
While this is in fact a unique review, the results from table are difficult to assimilate since the AIP values are high in several control studies, much higher that some patient studies. The reasons may be several, and it may be hard to distinguish since variations in the measurements and factors such as alcohol consumption, other not-prescription drug usage, ethnicity may play a role. Table also gives values of lipoproteins in some cases mg/dl and in some other cases mml/L. It is difficult to comprehend this table for lack of uniformity. I understand that the authors can only quote these studies and values, However, a uniformity in values provided in tables will help readers. since the HDL appears to be important in AIP values, variation in the measurement of HDL will cloud the results and conclusions. It is hard to conclude whether AIP can be taken as a risk factor assessment.
Some minor corrections. For example at 112 "familial" and not familiar.
Round 2
Reviewer 3 Report
The authors have revised the manuscript carefully and explained all of my issues. I have only one concern. In my opinion the authors should explain the meaning of question marks next to the WC and WHpR. The significance of "high?" and "low?" is unclear.
Author Response
Dear Reviewer,
Thank you for kindly reviewing our work for a second time and for your really helpful comments. We have tried to address your final query to the best of our ability as follows:
Point 1:
The authors have revised the manuscript carefully and explained all of my issues. I have only one concern. In my opinion the authors should explain the meaning of question marks next to the WC and WHpR. The significance of "high?" and "low?" is unclear.
Response to Point 1:
Many thanks for the comment and we are glad to hear that we have explained almost all your issues. Also, many thanks for highlighting your concern about the question marks. The reason for these was because in some studies gender may not have been reported and in these cases it was not possible to make a firm judgement about the reported index (since it may have been over the threshold for a male but not for a female etc.). We appreciate your point this being unclear and included an explanation of this in the results section (lines 183-185). We hope that this might improve the manuscript and make it clearer to you and other readers.
For more details please see the revised version manuscript.